# LATENT PDE MAPPING FOR SHAPE-GENERALIZABLE PHYSICS-INFORMED NEURAL NETWORKS

## ABSTRACT

Physics-Informed Neural Networks (PINNs) have shown strong potential for learning physically consistent representations from sparse data, but often struggle to generalize to geometries with varying shapes. To address this challenge, we introduce *latent PDE mapping*, a technique for mapping geometry-specific partial differential equations (PDEs) to a shared latent PDE representation using the deformation gradient. We embed latent PDE mapping into the PINN framework (LPM-PINN), enabling PINNs to capture geometric variability while preserving the governing physics. This integration facilitates accurate predictions of nonlinear, time-dependent systems even in geometries well beyond the training distribution. We demonstrate LPM-PINN on a challenging nonlinear time-dependent PDE with sharp gradients, the Aliev–Panfilov model of cardiac electrophysiology, in both 2D and 3D. Our results show that LPM-PINN generalizes robustly across diverse geometries, including shapes with drastically changing boundaries that lie outside the training distribution. These findings establish latent PDE mapping as a promising approach for boosting the geometric generalizability of physics-informed neural networks.

## 1 INTRODUCTION

Physics-informed neural networks (PINNs) (Raissi et al., 2019) have emerged as a new paradigm for learning physically consistent representations from sparse observations (Karniadakis et al., 2021; Cuomo et al., 2022). However, PINNs face significant challenges when making predictions on geometries with varying shapes, often requiring retraining when encountering novel morphologies outside the training distribution (Gao et al., 2021). This limitation is critical in time-sensitive applications (e.g. medicine) where short compute times and generalization across diverse physiologically-derived shapes are required. Here, we have chosen a prominent example that arises in cardiac electrophysiology, where accurate cardiac arrest risk assessments require adaptation to diverse heart geometries, and integration into medical workflows requires fast computations.

To address this issue, we introduce *latent PDE mapping*, a technique that maps geometry-specific partial differential equations (PDEs) to a shared latent PDE representation. Using affine shape parameterization, a predefined latent geometry, and the deformation gradient; our method expresses the loss terms of a PINN (LPM-PINN) using latent coordinates. This approach preserves the underlying dynamics while capturing geometric variability, enabling accurate predictions on unseen shapes in nonlinear, time-dependent systems. We apply latent PDE mapping to the Aliev-Panfilov model of cardiac electrophysiology, a representative benchmark for challenging nonlinear time-dependent PDE dynamics. The proposed approach offers a broadly applicable strategy for extending physics-informed neural networks to problems involving geometries with variable shapes, complementary to the current state of the art approaches involving operator and graph-neural architectures Li et al. (2023a;b); Yin et al. (2024); Zhong & Meidani (2025); Würth et al. (2024).

### 1.1 OUR CONTRIBUTIONS

- We introduce *latent PDE mapping*, a novel technique that maps geometry-specific PDEs to a shared latent PDE, enabling PINNs to learn meaningful representations from sparse observations across diverse geometrical shapes.

- We implement *latent PDE mapping* in a conceptually simple PINN framework (LPM-PINN), involving the Aliev–Panfilov model of cardiac electrophysiology, a challenging nonlinear, time-dependent PDE with sharp gradients in 2D and 3D. Our results show that LPM-PINN provides accurate solutions even in extreme rotation scenarios where the boundary changes radically.

- We provide theoretical and empirical evidence that latent PDE mapping properly accounts for geometric variability in the physics loss gradient, yielding more generalizable representations.

## 2 RELATED WORK

Recent developments within scientific machine learning have enabled the creation of flexible neural network PDE solvers that can generalize to new geometries without needing retraining. Neural operators Li et al. (2023a;b); Yin et al. (2024); Zhong & Meidani (2025), and graph neural networks Würth et al. (2024) are currently the two leading approaches. Neural operators possess rich mathematical universal approximation properties, guaranteeing that the neural network's parameterized solutions can approximate arbitrarily closely PDE solutions from varying geometries. Nevertheless, most neural operator approaches Li et al. (2023a;b); Yin et al. (2024) are data-hungry, often requiring extensive PDE solution datasets across diverse geometries to achieve high accuracy. This motivates the pursuit of data-efficient approaches capable of learning from fewer geometric samples, an essential consideration in domains where data collection is costly or ethically constrained, such as medicine. In response, PINNs have emerged, leveraging governing physical laws to learn effectively from sparse data Gao et al. (2021); Zhong & Meidani (2025); Würth et al. (2024); Dalton et al. (2023); Peng et al. (2023); Gao et al. (2022); Kashefi & Mukerji (2022).

A common approach for geometry-aware PINN studies has been to develop specialized network architectures, replacing multilayer perceptrons with physics-informed convolutional neural networks (Gao et al., 2021), physics-informed graph neural networks (Dalton et al., 2023; Peng et al., 2023; Würth et al., 2024; Gao et al., 2022), or physics-informed PointNet (Kashefi & Mukerji, 2022). These methods are better suited to handle variable geometries than basic fully-connected PINNs, but require uniform grids, complex meshing at inference, or struggle to generalize across PDE parameters (Zhong & Meidani, 2025). To overcome these challenges, PINNs have been augmented with shape descriptors (Regazzoni et al., 2022; Costabal et al., 2024) or global geometric parameters (Sun et al., 2023; Ghosh et al., 2024; Zhong & Meidani, 2025). While showing promising results, these methods formulate their physics losses in terms of the varying physical domains, which limits the gradient information available to the networks during training.

Another research direction involves combining physics-based losses with latent geometries, where inputs are embedded into a common latent space to facilitate comparison and efficient representation learning across different shapes. Regazzoni et al. (2022) proposed a universal latent space for parameterized geometries, enabling learning across varying shapes. Similarly, Mezzadri et al. (2023) introduced a framework that aligns geometric variability through latent embeddings, enabling simple linear elasticity models to generalize across freeform domains. More recently, Burbulla (2023) introduced a PDE mapping to low-dimensional manifolds and applied it to simple linear PDEs. However, the current latent-geometry PINN methods Mezzadri et al. (2023); Regazzoni et al. (2022); Burbulla (2023) are limited to simple, linear, static PDEs. This limits the methods' utility in real-world applications, which are often complex, nonlinear, and dynamic. Moreover, no work has yet shown that PINNs with mapped PDEs can generalize well to geometries outside of the training distribution.

Building on recent scientific machine learning studies involving mapped geometries Li et al. (2023a;b); Yin et al. (2024); Zhong & Meidani (2025); Mezzadri et al. (2023); Regazzoni et al. (2022); Burbulla (2023), the latent PDE mapping introduces a broadly applicable mathematical framework that moves beyond simple PDE mapping to PINN formulations with geometrically variable shapes and nonlinear, time-dependent PDEs. Furthermore, we introduce the use of the deformation gradient to accurately map nonlinear PDEs within PINNs, thereby enabling more accurate gradient calculations in which the effect of the geometric variability is included in the physics loss gradient.

## 3 LATENT PDE MAPPING

We consider a time-dependent PDE defined over a geometry $\Omega(s)$. Here, $s$ is a set of shape parameters describing the overall geometry of $\Omega$. The governing PDE is given as

$$\mathcal{F}\left(u\left(\boldsymbol{x},t;s\right)\right) = f(\boldsymbol{x},t,u;s), \qquad (\boldsymbol{x},t) \in \Omega(s) \times \mathcal{T} \tag{1}$$

where $\mathcal{F}$ denotes a differential operator, $f$ represents a source term that introduces external influences into the system, $\boldsymbol{x} \in \Omega(s) \subset \mathbb{R}^d$ are the spatial coordinates, $t \in \mathcal{T} \subset \mathbb{R}$ is the time, and $u$ is the unknown PDE solution. In practice, obtaining an exact solution to equation 1 is often intractable due to the complexity of the underlying system. To address this, we employ a PINN to approximate the solution such that

$$\mathcal{NN}(\boldsymbol{x},t,s;\theta) = u_\theta \approx u(\boldsymbol{x},t;s) \tag{2}$$

where $\theta$ represents the trainable parameters. PINNs are known to offer a data-efficient machine learning alternative by embedding physical laws directly into the neural network via PDE residuals in the loss function (Raissi et al., 2019). The residual is defined as

$$\mathcal{R} = \mathcal{F}\left(u\left(\boldsymbol{x},t;s\right)\right) - f(\boldsymbol{x},t,u;s) = 0, \qquad (\boldsymbol{x},t) \in \Omega(s) \times \mathcal{T} \tag{3}$$

where $\mathcal{R}$ depends on $\Omega(s)$ and the shape parameters $s$. With latent PDE mapping, we rather express the geometry-specific residual in equation 3 over a latent geometry. Thus, we assume that there exists a continuous map between $\Omega(s)$ and a predefined latent geometry $\Omega_0$, defined as

$$\Phi := \boldsymbol{X} \to \boldsymbol{x} \tag{4}$$

where $\boldsymbol{X}$ is a given point in $\Omega_0$ while $\boldsymbol{x}$ is the associated point in $\Omega(s)$. Physical quantities can be mapped from $\Omega(s)$ to $\Omega_0$, or vice versa, through the deformation gradient and deformation Jacobian given in their most general form as

$$\mathbf{F}(\boldsymbol{X},t,s) = \mathbf{I} + \nabla \boldsymbol{U}(\boldsymbol{X},t,s) \tag{5}$$

and $J(\boldsymbol{X},t,s) = \det(\mathbf{F})$, respectively (Holzaphel, 2000). Here, $\mathbf{I}$ is the identity tensor and $\boldsymbol{U}(\boldsymbol{X},t,s) = \boldsymbol{x}(\boldsymbol{X},t,s) - \boldsymbol{X}$ is the displacement field at time $t$ for the shape parameters $s$. In this study, we use the deformation gradient to map the geometry-specific $\mathcal{R}$ in equation 3 to a shared latent representation, yielding

$$\mathcal{R}(\boldsymbol{X},t,u,\mathbf{F},J) = \mathcal{F}\left(u\left(\boldsymbol{X},t;s\right),\mathbf{F},J\right) - f(\boldsymbol{X},t,u,\mathbf{F},J;s), \qquad (\boldsymbol{X},t) \in \Omega_0 \times \mathcal{T}. \tag{6}$$

In this way, the dependency on $s$ has been moved from the physical geometry $\Omega(s)$ into the PDE itself through the deformation gradient $\mathbf{F}$. This approach is what we refer to as the *latent PDE mapping* technique.

### 3.1 APPLICATION TO NONLINEAR, TIME-DEPENDENT, STIFF SYSTEMS: THE ALIEV-PANFILOV PDE

We demonstrate our latent PDE mapping technique on the Aliev-Panfilov model from cardiac eletrophysiology. The Aliev-Panfilov PDE (Aliev & Panfilov, 1996) is used to describe the evolution of transmembrane potential $V$ over a physical geometry representing cardiac tissue and offers a fair representation of challenging PDEs due to its nonlinearity, sharp gradients, and time-dependency. The PDE can be expressed over a physical geometry $\Omega(s)$ as

$$\begin{cases} \frac{\partial V}{\partial \tau} = \nabla \cdot (\mathbf{D}\nabla V) - kV(V-a)(V-1) - VW & \text{in } \Omega(s), \\ \frac{\partial W}{\partial \tau} = \left(\epsilon_0 + \frac{\mu_1 W}{V+\mu_2}\right)\left(-W - kV\left(V-a-1\right)\right) & \text{in } \Omega(s), \\ \mathbf{D}\nabla V \cdot \boldsymbol{n} = 0 & \text{on } \partial\Omega(s) \end{cases} \tag{7}$$

where $V, W$, and $\tau$ are dimensionless variables representing the transmembrane potential, recovery variable, and time, respectively. $V \in [0,1]$ is given in arbitrary units (AU), while $\tau = 12.9t$ is measured in temporal units (TU) with $t$ given in milliseconds. The tissue conductivity is defined by the diffusion tensor $\mathbf{D}$, while $k, a, \epsilon_0, \mu_1, \mu_2$ are parameters controlling the overall shape and temporal dynamics of $V$ and $W$. Additionally, the PDE employs a no-flux Neumann boundary condition where $\boldsymbol{n}$ is the vector normal to the boundary of $\Omega(s)$. Consequently, there is no leakage of $V$ to regions outside of $\Omega(s)$.

We apply our latent PDE mapping technique to the Aliev-Panfilov PDE in equation 7. For a time-independent mapping, the latent PDE representation is given as

$$
\begin{cases}
\frac{\partial V}{\partial \tau} = \frac{1}{J}\nabla \cdot (J\mathbf{F}^{-1}\mathbf{D}\mathbf{F}^{-T}\nabla V) - kV(V-a)(V-1) - VW & \text{in } \Omega_0, \\
\frac{\partial W}{\partial \tau} = \left(\epsilon_0 + \frac{\mu_1 W}{V+\mu_2}\right)(V,W)\left(-W - kV\left(V-a-1\right)\right) & \text{in } \Omega_0, \\
J\mathbf{F}^{-1}\mathbf{D}\mathbf{F}^{-T}\nabla V \cdot \boldsymbol{N} = 0 & \text{on } \partial\Omega_0,
\end{cases}
\tag{8}
$$

where $\boldsymbol{N}$ is the normal vector to the boundary of the latent geometry $\Omega_0$, $\mathbf{F} = \mathbf{F}(\boldsymbol{X}, s)$ and $J = J(\boldsymbol{X}, s)$. A detailed derivation of equation 8 can be found in Appendix A.

## 3.2 ACCURATE GRADIENT CALCULATION WITH LATENT PDE MAPPING

The physics loss in PINNs is typically evaluated with the mean squared error (MSE) of $\mathcal{R}$ (Wang et al., 2023), given as

$$
\mathcal{L}_{phys} = \frac{1}{N_{phys}}\sum_i^{N_{phys}} \mathcal{R}_i^2
\tag{9}
$$

using a traditional mini-batch approach with $N_{phys}$ collocation points to evaluate $\mathcal{R}$. This approach treats $\mathcal{R}$ as independent of $\Omega(s)$ during optimization, which is not the case and can lead to inaccurate gradient estimates. Thus, a more accurate formulation is to evaluate the physics loss as a continuous integral

$$
\mathcal{L}_{phys} = \int_{\Omega(s)} \mathcal{R}(\boldsymbol{x}, t, u, s)^2 \mathrm{d}\Omega
\tag{10}
$$

and apply the Leibniz integral rule when computing the shape gradient $\frac{\partial \mathcal{L}_{phys}}{\partial s}$. This results in

$$
\frac{\partial \mathcal{L}_{phys}}{\partial s} = \int_{\Omega(s)} \frac{\partial}{\partial s}\mathcal{R}(\boldsymbol{x}, t, u, s)^2 \mathrm{d}\Omega + \int_{\partial\Omega(s)} \mathcal{R}(\boldsymbol{x}, t, u, s)^2 \frac{\partial \boldsymbol{x}}{\partial s}\cdot \boldsymbol{n}\mathrm{d}S
\tag{11}
$$

where $\boldsymbol{n}$ is the outward unit normal to the boundary $\partial\Omega(s)$ and $\mathrm{d}S$ is an infinitesimally small part of the boundary. The second term in equation 11 accounts for the movement of the boundary, which is neglected in the discrete loss formulation in equation 9. This omission can lead to inaccurate gradient estimates, hindering training and resulting in suboptimal PINNs.

With latent PDE mapping, the dependency of $s$ is moved from the geometry into the PDE itself via the deformation gradient $\mathbf{F}$. Consequently, the integrand does not vary with $s$ and the shape gradient can be computed directly

$$
\frac{\partial \mathcal{L}_{phys}}{\partial s} = \int_{\Omega_0} \frac{\partial}{\partial s}\mathcal{R}(\boldsymbol{X}, t, u, \mathbf{F}, J)^2 \mathrm{d}\Omega_0.
\tag{12}
$$

Thus, the straightforward MSE in equation 9 can be applied during training without sacrificing gradient accuracy. Based on these considerations, we hypothesized that improving the accuracy of the physics loss gradient via latent PDE mapping can improve the generalizability of PINNs to novel geometries.

## 4 CARDIAC ELECTROPHYSIOLOGY DATASETS WITH VARIABLE GEOMETRIES

We constructed four and seven families of geometries in 2D and 3D, respectively, for training and testing of our PINNs. In 2D, the latent geometry $\Omega_0$ was defined as a $10\times10$ mm square, while in 3D it was a $10 \times 10 \times 10$ mm cube. Families belonging to 2D and 3D are denoted with a $\mathcal{G}$ and $\mathcal{H}$, respectively. In the following, we describe the generation of 2D datasets. The extension to 3D is straightforward and provided in detail in Appendix B.2.

The geometries were generated by deforming $\Omega_0$ through different affine transformations expressed in their most general form as

$$
\boldsymbol{x} = \boldsymbol{A}\boldsymbol{X} + \boldsymbol{X}^T\boldsymbol{M}\boldsymbol{X}
\tag{13}
$$

with

$$\boldsymbol{A} = \begin{bmatrix} a_1 & a_2 \\ a_3 & a_4 \end{bmatrix}, \qquad \boldsymbol{M} = \begin{bmatrix} m_1 & 0 \\ 0 & m_4 \end{bmatrix}. \tag{14}$$

The elements of $\boldsymbol{A}$ and $\boldsymbol{M}$, referred to as *affine parameters*, are denoted by $s = \{a_1, a_2, a_3, a_4, m_1, m_4\}$. Each family of geometries corresponded to a distinct deformation type: expansion ($\mathcal{G}_{exp}$), shearing ($\mathcal{G}_{shear}$), nonlinear deformation ($\mathcal{G}_{nonlin}$), and rotation ($\mathcal{G}_{rot}$). Figure 1 illustrates one representative geometry from each family, while Table 5 in Appendix B.1 gives the affine parameter ranges for all families. The deformations employed in this work were static in time; however, the approach can be extended to time-dependent deformations as shown in Section 3.

For each family, we generated two branches. The first branch ($\mathcal{G}_k$) contained 50 geometries, which were later split into training, validation, and test sets. The second branch ($\mathcal{G}_k^*$) contained 35 geometries generated from parameter ranges outside those of $\mathcal{G}_k$, and was used exclusively for testing. We refer to test geometries in $\mathcal{G}_k$ as the *internal family* and $\mathcal{G}_k^*$ as the *external family*.

In 3D, the same procedure was applied with the same number of geometries and versions per family. However, only linear deformation types were considered. The 3D families are denoted as $\mathcal{H}_k^p$ where $k$ indicates the deformation type (expansion, shearing, or rotation) and $p$ indicates the direction of the deformation when applicable ($\mathcal{H}_{rot}^x$ = rotation about the $x$-axis, $\mathcal{H}_{shear}^{xy}$ = shearing along the $xy$-plane, etc.).

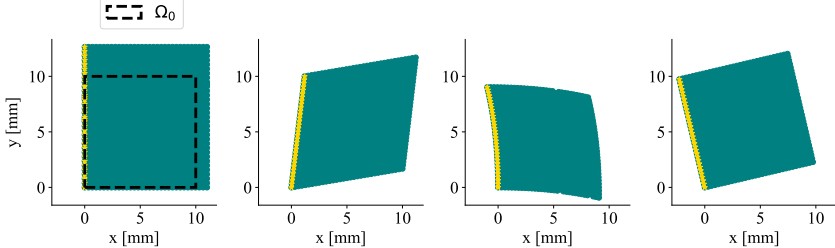

Figure 1: From left to right, the figure shows an example of a geometry from $\mathcal{G}_{exp}$, $\mathcal{G}_{shear}$, $\mathcal{G}_{nonlin}$, and $\mathcal{G}_{rot}$. All geometries were externally stimulated at the left edge nodes (yellow) in the isotropic scenario. The dashed line illustrates the latent geometry $\Omega_0$ in 2D.

**Synthetic cardiac electrophysiology data**   We used *openCARP* (Plank* et al., 2021; openCARP consortium et al., 2024) to create synthetic data that was used to approximate the ground truth PDE solution during training and testing of the PINNs. Thus, we solved the Aliev-Panfilov PDE in equation 7 over the physical geometries using the finite element method (FEM). We created both isotropic and anisotropic datasets to explore different PDE dynamics. In the isotropic case, all geometries were stimulated by an external current at nodes located at the left boundary/plane. Sheet fibers were oriented along the $x$-axis with, resulting in a planar wave propagation. The anisotropic datasets were generated by applying a point stimulus to all nodes within a radius of 0.75 mm in the center of the geometry. The fiber orientations were deformed according to the affine transformation to ensure consistent PDE dynamics. In both cases, all simulations were run for 520 ms, yielding a full cycle of polarization and re-polarization. Solutions at $t < 6\,\text{ms}$ were excluded to remove applied current from the system. The exact configurations used for synthetic data generation are listed in Table 8 and 9 in Appendix D.

## 5   PINN TRAINING PROCEDURES AND EVALUATION METHODS

We developed three PINNs for comparison and evaluation of the latent PDE mapping. The first PINN (LPM-PINN) incorporates the latent PDE mapping technique and uses affine parameters as additional inputs. The second (Affine-PINN) and third (Basic-PINN) PINNs adopt the conventional physics loss formulated over the physical geometries. However, the Affine-PINN integrates affine parameters as input, whereas the Basic-PINN relies exclusively on spatiotemporal inputs. This setup enables a systematic evaluation of the contribution of latent PDE mapping to PINN training, as well as the added benefits of including affine parameters.

All PINNs were trained by minimizing a hybrid loss function defined as

$$\mathcal{L}(\theta) = \mathcal{L}_{data}(\theta) + \mathcal{L}_{phys}(\theta) + \mathcal{L}_{bc}(\theta) + \mathcal{L}_{ic}(\theta) \tag{15}$$

where $\mathcal{L}_{data}(\theta)$ is the loss due to known FEM data, $\mathcal{L}_{phys}(\theta)$ is the loss described by the governing PDE residual, $\mathcal{L}_{bc}(\theta)$ is the loss associated with the boundary condition, and $\mathcal{L}_{ic}(\theta)$ is the loss associated with the initial condition. The loss terms were equally weighted, and each term was evaluated using the MSE over a given set of spatiotemporal points ($\mathcal{N}_{data}, \mathcal{N}_{phys}, \mathcal{N}_{bc}, \mathcal{N}_{ic}$). Furthermore, we defined the physics and boundary loss in LPM-PINN as

$$\mathcal{L}_{phys}^{LPM} \equiv \frac{1}{\mathcal{N}_{phys}} \sum^{\mathcal{N}_{phys}} \mathcal{R}(\boldsymbol{X}, \tau, \hat{V}, \hat{W}, \mathbf{F}, J), \qquad \mathcal{L}_{bc}^{LPM} \equiv \frac{1}{\mathcal{N}_{bc}} \sum^{\mathcal{N}_{bc}} \mathcal{R}(\boldsymbol{X}, \tau, \hat{V}, \mathbf{F}, J) \tag{16}$$

and the conventional losses in Affine-PINN and Basic-PINN as

$$\mathcal{L}_{phys}^{conv} \equiv \frac{1}{\mathcal{N}_{phys}} \sum^{\mathcal{N}_{phys}} \mathcal{R}(\boldsymbol{x}, \tau, \hat{V}, \hat{W}), \qquad \mathcal{L}_{bc}^{conv} \equiv \frac{1}{\mathcal{N}_{bc}} \sum^{\mathcal{N}_{bc}} \mathcal{R}(\boldsymbol{x}, \tau, \hat{V}). \tag{17}$$

where $\boldsymbol{X} \in \Omega_0$ and $\boldsymbol{x} \in \Omega(s)$. For the isotropic datasets, each PINN consisted of a fully connected neural network with 10 hidden layers with 25 neurons in each layer, while for the anisotropic datasets, each PINN had 8 hidden layers with 64 neurons each to reflect the increased complexity of the PDE dynamics. Furthermore, we employed the *tanh* as activation function in all cases to handle second-order derivatives (equation 8) needed to calculate the physics loss (equation 9). All PINNs predicted $\hat{V}$ and $\hat{W}$ as outputs. A complete overview of the hyperparameters for each PINN can be found in Table 7 in Appendix C.

Each internal family ($\mathcal{G}_k, \mathcal{H}_k$) was split into a training set, validation set, and test set. Unlike the conventional split used in machine learning, we adopted an inverted allocation strategy with 20% train data, 10% validation data, and 70% test data in order to restrict the available training data. Thus, each family ($\mathcal{G}_k, \mathcal{H}_k$) had 10 train geometries, 5 validation geometries, and 35 test geometries. Additionally, in some experiments, we merged two families to generate a dataset ($\mathcal{G}_{k1} + \mathcal{G}_{k2}$) with greater geometric variability. In these cases, each family contributed equally to each split, resulting in 20 train geometries, 10 validation geometries, and 70 test geometries. Furthermore, we selected $\mathcal{N}_{data} = 14$, $\mathcal{N}_{phys} = 700$, $\mathcal{N}_{bc} = 80$, and $\mathcal{N}_{ic} = 30$ spatial locations from each geometry in the training set and trained the models for 5000 epochs. $\mathcal{N}_{phys}$, $\mathcal{N}_{bc}$, and $\mathcal{N}_{ic}$ were resampled at every epoch to ensure that the physics was learned over the entire geometry.

During training, we evaluated $\mathcal{L}(\theta)$ for each geometry in the validation set. Since our validation set spanned multiple distinct geometries, we selected the best PINN state as the state that gave the lowest maximum $\mathcal{L}(\theta)$ across the validation geometries, rather than the lowest average $\mathcal{L}(\theta)$. This criterion ensured that the PINN generalized effectively to geometries differing substantially from those seen during training. To reduce computational overhead, we computed the validation loss every 10 epochs using a subsample of points from each geometry.

**Evaluation metrics** We employed the relative $L_2$ error ($\varepsilon_{L2}$) as an evaluation metric, given as

$$\varepsilon_{L2} = \frac{\sqrt{\sum_i^{N_{test}} \left(\hat{V}_i - V_i\right)^2}}{\sqrt{\sum_i^{N_{test}} V_i^2}} \tag{18}$$

where $\hat{V}$ is the predicted transmembrane potential and $V$ is the approximated FEM data used as ground truth. Results are presented as the mean relative $L_2$ error across all geometries in the given family with the corresponding standard deviation.

## 6  EXPERIMENTS

In the following sections, we present results from a series of experiments used to evaluate and compare the PINNs' performance when generalizing across diverse geometries in 2D and 3D. Furthermore, we investigate the role of the missing boundary shape gradients when latent PDE mapping is not applied.

## 6.1 DOES LATENT PDE MAPPING IMPROVE GEOMETRIC GENERALIZABILITY IN 2D?

The results indicate consistently low prediction errors for all PINNs across test geometries in the internal families $\mathcal{G}_{exp}$, $\mathcal{G}_{shear}$, and $\mathcal{G}_{nonlin}$ when applied to isotropic PDE dynamics (Table 1). Moreover, the results show that LPM-PINN and Affine-PINN generalize to the corresponding external families with only a modest increase in prediction error, whereas the Basic-PINN exhibits errors of an order of magnitude higher on the same families. Notably, LPM-PINN is the only PINN that achieves accurate predictions on the $\mathcal{G}_{rot}^*$ family, while Affine-PINN and Basic-PINN yield significantly inaccurate results, as illustrated in the last row of Figure 5 in Appendix E.1. These findings demonstrate that latent PDE mapping improves geometric generalizability, particularly when the boundary undergoes radical changes.

Table 2 shows that LPM-PINN can learn and make accurate predictions when trained on geometries from two different families with isotropic PDE dynamics. In contrast, Affine-PINN and Basic-PINN fail to learn meaningful representations in the same setting, except for Affine-PINN on $\mathcal{G}_{shear} + \mathcal{G}_{rot}$. Figure 6 in Appendix E.1 visualizes predictions on the same geometries as in Figure 5 in Appendix E.1, showing that the higher error is not limited to the $\mathcal{G}_{rot}$ and $\mathcal{G}_{rot}^*$ family, but arises from degraded performance across both families. Hence, the results indicate that latent PDE mapping can enhance generalizability when learning across fundamentally different geometries.

Table 1: Mean relative $L_2$ PINN-FEM discrepancy $\pm$ std evaluated over the internal ($\mathcal{G}_k$) and external ($\mathcal{G}_k^*$) test geometries of each geometry family in 2D isotropic scenarios.

|  | LPM-PINN | Affine-PINN | Basic-PINN |
|---|---|---|---|
| $\mathcal{G}_{exp}$ | **0.019 ± 0.004** | 0.024 ± 0.005 | 0.057 ± 0.028 |
| $\mathcal{G}_{exp}^*$ | **0.044 ± 0.013** | 0.070 ± 0.036 | 0.166 ± 0.043 |
| $\mathcal{G}_{shear}$ | **0.024 ± 0.002** | 0.029 ± 0.004 | 0.082 ± 0.029 |
| $\mathcal{G}_{shear}^*$ | **0.074 ± 0.027** | 0.077 ± 0.024 | 0.203 ± 0.041 |
| $\mathcal{G}_{nonlin}$ | **0.029 ± 0.005** | 0.029 ± 0.009 | 0.058 ± 0.024 |
| $\mathcal{G}_{nonlin}^*$ | 0.055 ± 0.020 | **0.054 ± 0.019** | 0.126 ± 0.044 |
| $\mathcal{G}_{rot}$ | **0.017 ± 0.001** | 0.055 ± 0.016 | 0.229 ± 0.021 |
| $\mathcal{G}_{rot}^*$ | **0.020 ± 0.002** | 0.272 ± 0.137 | 0.331 ± 0.042 |

**Does latent PDE mapping handle anisotropic PDE dynamics?** Table 3 shows that both LPM-PINN and Affine-PINN make accurate predictions on internal test geometries, whereas the Basic-PINN struggles with anisotropic PDE dynamics. The table and visualization in Figure 2 also indicate that LPM-PINN generalizes better to external geometries than Affine-PINN, suggesting that it learns a more robust representation of the anisotropic dynamics. The results demonstrate that as the complexity of the underlying problem increases, the benefits of an explicit latent representation become more pronounced.

## 6.2 DOES LATENT PDE MAPPING IMPROVE GEOMETRIC GENERALIZABILITY IN 3D?

The results show that all PINNs can generalize to both the internal and external families when tested on rotations around the $x$-axis and on shearing along the $yz$-plane ($\mathcal{H}_{rot}^x$, $\mathcal{H}_{rot}^{x*}$, $\mathcal{H}_{shear}^{yz}$ and $\mathcal{H}_{shear}^{yz*}$; Table 4). Beyond these settings, LPM-PINN and Affine-PINN generalize well to the remaining shearing directions and expansion families, whereas Basic-PINN struggles to make accurate predictions on the corresponding external families (Figure 7 in Appendix E.2). Table 4 further shows that LPM-PINN is the only PINN capable of handling rotations around the $y$- and $z$-axes. In these cases, both Affine-PINN and Basic-PINN produce entirely inaccurate predictions on the external families, as illustrated in Figure 3.

## 6.3 HOW LARGE ARE THE MISSING BOUNDARY SHAPE GRADIENTS WHEN LATENT PDE MAPPING IS NOT USED?

Figure 4 shows that the omitted boundary information (equation 11) in the shape gradients is large across all 2D cases (see Appendix F for computational details). For every family shown in Figure 4, the missing boundary information (blue) exceeds the shape gradient used in the Affine-PINN

Table 2: Mean relative $L_2$ PINN-FEM discrepancy $\pm$ std evaluated over the internal ($\mathcal{G}_k$) and external ($\mathcal{G}_k^*$) test geometries from a combination of families in 2D isotropic scenarios.

|  | LPM-PINN | Affine-PINN | Basic-PINN |
|---|---|---|---|
| $\mathcal{G}_{exp}$+ $\mathcal{G}_{rot}$ | **0.022 $\pm$ 0.006** | 0.911 $\pm$ 0.065 | 0.199 $\pm$ 0.061 |
| $\mathcal{G}_{exp}^*$+ $\mathcal{G}_{rot}^*$ | **0.138 $\pm$ 0.102** | 1.111 $\pm$ 0.268 | 0.285 $\pm$ 0.084 |
| $\mathcal{G}_{shear}$+ $\mathcal{G}_{rot}$ | **0.021 $\pm$ 0.005** | 0.027 $\pm$ 0.003 | 2.571 $\pm$ 0.067 |
| $\mathcal{G}_{shear}^*$+ $\mathcal{G}_{rot}^*$ | **0.048 $\pm$ 0.033** | 0.152 $\pm$ 0.128 | 2.634 $\pm$ 0.174 |
| $\mathcal{G}_{nonlin}$+ $\mathcal{G}_{rot}$ | **0.024 $\pm$ 0.003** | 1.104 $\pm$ 0.114 | 0.186 $\pm$ 0.073 |
| $\mathcal{G}_{nonlin}^*$+ $\mathcal{G}_{rot}^*$ | **0.035 $\pm$ 0.013** | 1.181 $\pm$ 0.181 | 0.272 $\pm$ 0.122 |

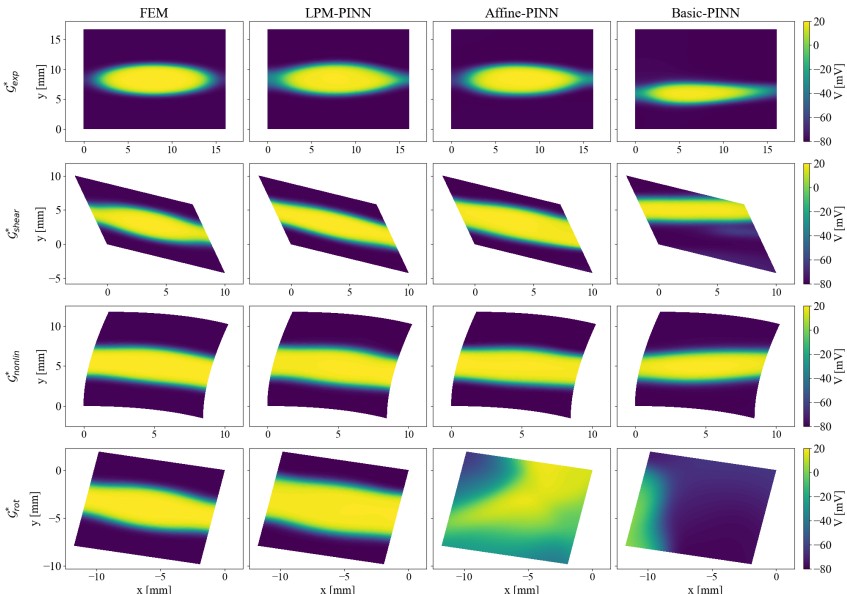

Figure 2: Snapshots of predicted transmembrane voltages ($V$) at $t = 50$ ms in the anisotropic scenario. Each row corresponds to a geometry taken from the presented external family ($\mathcal{G}_{exp}^*$, $\mathcal{G}_{shear}^*$, $\mathcal{G}_{nonlin}^*$, $\mathcal{G}_{rot}^*$). The left column shows the FEM ground truth approximation.

(orange). The magnitude of this missing information depends on the boundary movement when making changes to $s$ (Figure 10a in Appendix F): the family with the smallest gap in Figure 4 also exhibits the smallest boundary change in Figure 10a. A similar trend appears in 3D, where shearing families show the lowest boundary changes (Figure 10b, Appendix F) and correspondingly small missing information (Figure 8, Appendix E.2), while expansion and rotational families exhibit larger boundary changes (Figure 10b, Appendix F) and larger omissions (Figure 9, Appendix E.2). These findings indicate that the missing boundary shape gradients are of substantial sizes when latent PDE mapping is not applied, leading to suboptimal learning.

## 7 DISCUSSION

This work presents *latent PDE mapping*, a novel technique for mapping geometry-specific PDEs to a shared latent PDE representation. Latent PDE mapping moves the shape dependence from the geometry into the PDE itself through the deformation gradient. This representation allows essential boundary information to be incorporated into the physics loss during training of PINNs (LPM-PINN).

The empirical results demonstrate that latent PDE mapping enhances generalization across diverse 2D and 3D geometries for both isotropic and anisotropic PDE dynamics. In particular, the method is advantageous in scenarios where the training data comprises multiple geometric types (Table 2,

Table 3: Mean relative $L_2$ PINN-FEM discrepancy $\pm$ std evaluated over the internal ($\mathcal{G}_k$) and external ($\mathcal{G}_k^*$) test geometries of each geometry family in 2D anisotropic scenarios.

| | LPM-PINN | Affine-PINN | Basic-PINN |
|---|---|---|---|
| $\mathcal{G}_{exp}$ | $0.040 \pm 0.010$ | $\mathbf{0.038 \pm 0.008}$ | $0.205 \pm 0.062$ |
| $\mathcal{G}_{exp}^*$ | $\mathbf{0.071 \pm 0.017}$ | $0.074 \pm 0.013$ | $0.496 \pm 0.105$ |
| $\mathcal{G}_{shear}$ | $\mathbf{0.053 \pm 0.016}$ | $0.061 \pm 0.017$ | $0.229 \pm 0.066$ |
| $\mathcal{G}_{shear}^*$ | $0.125 \pm 0.072$ | $\mathbf{0.125 \pm 0.037}$ | $0.444 \pm 0.063$ |
| $\mathcal{G}_{nonlin}$ | $\mathbf{0.062 \pm 0.021}$ | $0.065 \pm 0.020$ | $0.151 \pm 0.037$ |
| $\mathcal{G}_{nonlin}^*$ | $\mathbf{0.108 \pm 0.037}$ | $0.115 \pm 0.037$ | $0.266 \pm 0.078$ |
| $\mathcal{G}_{rot}$ | $\mathbf{0.052 \pm 0.020}$ | $0.102 \pm 0.047$ | $0.420 \pm 0.031$ |
| $\mathcal{G}_{rot}^*$ | $\mathbf{0.180 \pm 0.094}$ | $0.650 \pm 0.181$ | $0.582 \pm 0.098$ |

Table 4: Mean relative $L_2$ FEM-PINN discrepancy $\pm$ std evaluated over the internal ($\mathcal{H}_k^p$) and external ($\mathcal{H}_k^{p*}$) test geometries from the geometry families in 3D isotropic scenarios.

| | LPM-PINN | Affine-PINN | Basic-PINN |
|---|---|---|---|
| $\mathcal{H}_{exp}$ | $\mathbf{0.015 \pm 0.001}$ | $0.015 \pm 0.002$ | $0.047 \pm 0.019$ |
| $\mathcal{H}_{exp}^*$ | $\mathbf{0.050 \pm 0.017}$ | $0.082 \pm 0.025$ | $0.166 \pm 0.044$ |
| $\mathcal{H}_{shear}^{xy}$ | $\mathbf{0.020 \pm 0.006}$ | $0.023 \pm 0.004$ | $0.083 \pm 0.027$ |
| $\mathcal{H}_{shear}^{xy*}$ | $0.077 \pm 0.030$ | $\mathbf{0.075 \pm 0.020}$ | $0.212 \pm 0.049$ |
| $\mathcal{H}_{shear}^{xz}$ | $\mathbf{0.020 \pm 0.005}$ | $0.024 \pm 0.004$ | $0.078 \pm 0.025$ |
| $\mathcal{H}_{shear}^{xz*}$ | $0.072 \pm 0.029$ | $\mathbf{0.068 \pm 0.017}$ | $0.209 \pm 0.049$ |
| $\mathcal{H}_{shear}^{yz}$ | $0.015 \pm 0.002$ | $\mathbf{0.014 \pm 0.001}$ | $0.016 \pm 0.000$ |
| $\mathcal{H}_{shear}^{yz*}$ | $0.018 \pm 0.005$ | $0.017 \pm 0.003$ | $\mathbf{0.016 \pm 0.001}$ |
| $\mathcal{H}_{rot}^{x}$ | $0.016 \pm 0.004$ | $0.021 \pm 0.011$ | $\mathbf{0.015 \pm 0.001}$ |
| $\mathcal{H}_{rot}^{x*}$ | $0.070 \pm 0.034$ | $0.073 \pm 0.035$ | $\mathbf{0.020 \pm 0.004}$ |
| $\mathcal{H}_{rot}^{y}$ | $\mathbf{0.014 \pm 0.001}$ | $2.014 \pm 0.188$ | $0.234 \pm 0.057$ |
| $\mathcal{H}_{rot}^{y*}$ | $\mathbf{0.036 \pm 0.023}$ | $1.212 \pm 0.134$ | $0.369 \pm 0.094$ |
| $\mathcal{H}_{rot}^{z}$ | $\mathbf{0.012 \pm 0.000}$ | $0.033 \pm 0.006$ | $0.224 \pm 0.050$ |
| $\mathcal{H}_{rot}^{z*}$ | $\mathbf{0.014 \pm 0.002}$ | $0.402 \pm 0.182$ | $0.382 \pm 0.074$ |

Figure 6) or where boundary conditions undergo significant variation due to rotations (last row in Figure 2 and 3). In such settings, conventional PINNs that rely exclusively on shape descriptors exhibit reduced performance, while latent PDE mapping provides a more robust learning representation.

A central insight emerging from this study concerns the role of missing boundary shape gradients. Adding the boundary gradient via latent PDE mapping can boost the ability of PINNs to generalize to new shapes (see LPM-PINN versus Affine-PINN in Table 1-4). Indeed, in the absence of latent PDE mapping, the omitted boundary terms can be larger than the remaining gradients (Figure 4). However, boundary gradient size does not necessarily translate directly into performance improvement. Thus, there is a need for more research to further investigate this issue.

The utility of latent PDE mapping depends on the overall boundary movement in the geometric families and on the PDE dynamics. Families with high boundary movements have a correspondingly higher missing boundary shape gradient when latent PDE mapping is not applied. Furthermore, latent PDE mapping improves learning and generalizability when geometric variability modifies the underlying PDE dynamics. In our case, the results show strong improvements when the initial activation site is moved substantially (last row in Figure 2 and 3).

It should be noted that latent PDE mapping introduces an additional computational overhead. As shown in Tables 11 and 12 in Appendix G, the training and inference times for LPM-PINN and Affine-PINN are largely comparable. However, the mapping to the reference geometry and deformation gradient computation add extra preprocessing costs with an average cost of $4.59 \pm 1.06$ seconds per geometry in 2D and an average cost of $35.01 \pm 2.33$ seconds per geometry in 3D. Importantly, this overhead is incurred only once prior to training or inference. Thus, the improvement

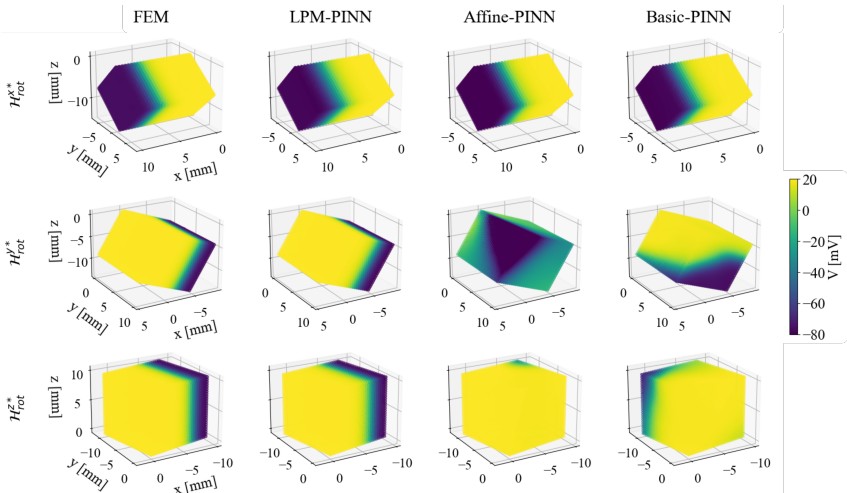

Figure 3: Snapshots of predicted transmembrane voltages ($V$) at $t = 50$ ms. Each row corresponds to a geometry taken from the presented external family ($\mathcal{H}_{rot}^{x*}$, $\mathcal{H}_{rot}^{y*}$, $\mathcal{H}_{rot}^{z*}$). The left column shows the FEM ground truth approximation.

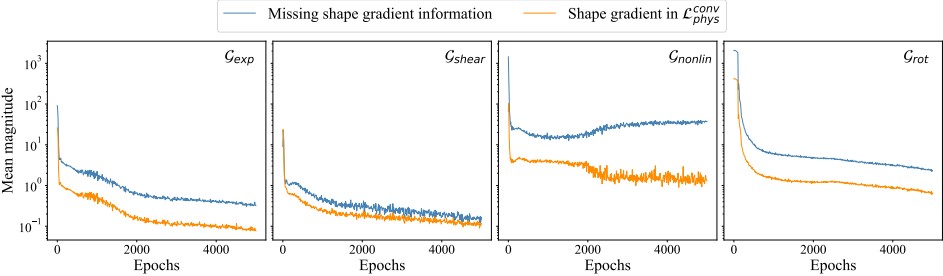

Figure 4: Numerical approximations of the missing shape gradients at the boundary and shape gradients used in $\mathcal{L}_{phys}^{conv}$ when training the Affine-PINN. The figure shows the mean magnitude across training geometries in $\mathcal{G}_{exp}$, $\mathcal{G}_{shear}$, $\mathcal{G}_{nonlin}$, and $\mathcal{G}_{rot}$.

in external predictive performance offered by LPM-PINN must be weighed against this additional data-preparation step. In practice, this cost is modest: the data mapping and deformation gradient computation were performed on a laptop CPU (Intel Core Ultra 9 185H) for our experiments, and could be significantly reduced by offloading these operations to a GPU.

**Limitations and future work** Our study has several limitations that open directions for future research. First, we relied on parameterized geometries, which may not always be available for more complex or realistic geometries encountered in real-world applications. Thus, extending latent PDE mapping to alternative shape representations is a critical future direction. One possibility is to employ principal component analysis modes as inputs to the PINN, rather than affine parameters, which has shown promise for representing cardiac geometries (Yin et al., 2024; Mauger et al., 2019). While our preliminary experiments (Table 10 in Appendix E.1) suggest that such extensions are feasible, a comprehensive exploration is beyond the scope of this paper. Second, the current study focuses exclusively on the Aliev–Panfilov model. Although the latent PDE mapping technique is, in principle, applicable to a broad class of architectures and physical systems, its use in alternative PDE settings remains an open direction for future work. Finally, our validation of the advantage of latent PDE mapping was limited to rotation transformations and simple geometries. The effectiveness of latent PDE mapping in more complex geometries remains to be determined. This will be an essential next step for proving the applicability of latent PDE mapping in realistic industrial and medical scenarios.

REPRODUCIBILITY STATEMENT

Synthetic datasets can be created by following the description given in Section 4 and Appendix D with parameter ranges as presented in Table 5 in Appendix B.1 for 2D and Table 6 in Appendix B.2 for 3D. Implementation details regarding developed PINNs are presented in Section 5 and Appendix C, where the selected hyperparameters for each PINN are presented in Table 7. The source code and datasets used to reproduce results in Section 6 will be shared in the camera-ready submission, if accepted, to preserve anonymity during the double-blind review process.

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

## A  DERIVATION OF LATENT PDE MAPPING APPLIED TO THE ALIEV-PANFILOV PDE

In the following section, we give a detailed derivation of how the Aliev-Panfilov PDE in equation 7 is mapped from a physical geometry $\Omega(s)$ to a latent geometry $\Omega_0$. For convenience, we restate the equations over $\Omega(s)$ here as

$$\begin{cases} \frac{\partial V}{\partial \tau} = \nabla \cdot (\mathbf{D}\nabla V) - kV(V-a)(V-1) - VW & \text{in } \Omega(s), \\ \frac{\partial W}{\partial \tau} = \left(\epsilon_0 + \frac{\mu_1 W}{V+\mu_2}\right)(-W - kV(V-a-1)) & \text{in } \Omega(s), \\ \mathbf{D}\nabla V \cdot \boldsymbol{n} = 0 & \text{on } \partial\Omega(s). \end{cases} \quad (19)$$

The mapping is achieved by applying the deformation gradient $\mathbf{F}(\boldsymbol{X}, t, s)$ and the deformation Jacobian $J(\boldsymbol{X}, t, s)$ to quantities in equation 19, as well as performing a variable substitution $\boldsymbol{x} \to \boldsymbol{X}$ where $\boldsymbol{x} \in \Omega(s)$ and $\boldsymbol{X} \in \Omega_0$. The deformation gradient $\mathbf{F}$ is given as

$$\mathbf{F}(\boldsymbol{X}, t, s) = \mathbf{I} + \nabla \boldsymbol{U}(\boldsymbol{X}, t, s) \quad (20)$$

where $\boldsymbol{U}(\boldsymbol{X}, t, s) = \boldsymbol{x}(\boldsymbol{X}, t, s) - \boldsymbol{X}$, while the deformation Jacobian is given as $J(\boldsymbol{X}, t, s) = \det(\mathbf{F})$.

We start by introducing how quantities in equation 19 are mapped when applying principles from nonlinear solid mechanics (Holzaphel, 2000). Quantities that do not involve any divergences or gradients are mapped directly through a volume change defined as

$$\mathrm{d}v = J\mathrm{d}V \quad (21)$$

where $dv$ and $dV$ are infinitesimally small volume elements in $\Omega(s)$ and $\Omega_0$, respectively. Gradients of a scalar field $\phi$ are mapped as

$$\nabla\phi(\boldsymbol{x}, t) = \mathbf{F}^{-T}\nabla\phi(\boldsymbol{X}, t) \quad (22)$$

which is obtained by applying the chain rule to $\nabla\phi(\boldsymbol{x}, t)$. Finally, Nanson's formula is used to map vector elements from $\Omega(s)$ to $\Omega_0$. The formula yields

$$\mathrm{d}s\boldsymbol{n} = J\mathbf{F}^{-T}\mathrm{d}S\boldsymbol{N} \quad (23)$$

where $\mathrm{d}s\boldsymbol{n}$ and $\mathrm{d}S\boldsymbol{N}$ give the vector elements of infinitesimally small surface areas defined on $\Omega(s)$ and $\Omega_0$.

Next, we rewrite the divergence term in equation 19 in integral form as

$$\int_{\Omega(s)} \nabla \cdot (\mathbf{D}\nabla V)\mathrm{d}\Omega$$

By applying Gauss's divergence theorem, we have that

$$\int_{\Omega(s)} \nabla \cdot (\mathbf{D}\nabla V)\mathrm{d}\Omega = \int_{\partial\Omega(s)} \mathbf{D}\nabla V \cdot \mathrm{d}s\boldsymbol{n} \quad (24)$$

where $\partial\Omega(s)$ is the surface of $\Omega(s)$ and $\boldsymbol{n}$ is the vector normal to the surface. We utilize the relationship of gradients in equation 22 and Nanson's formula in equation 23, such that the divergence term in equation 24 can be expressed over $\Omega_0$ as

$$\int_{\partial\Omega(s)} \mathbf{D}\nabla V \cdot \mathrm{d}s\boldsymbol{n} = \int_{\partial\Omega_0} \mathbf{D}\mathbf{F}^{-T}\nabla V \cdot J\mathbf{F}^{-T}\mathrm{d}S\boldsymbol{N} \quad (25)$$

In the 2D case, we have that

$$\mathbf{D} \in \mathbb{R}^{2x2}, \qquad \mathbf{F} \in \mathbb{R}^{2x2}, \qquad \nabla V \in \mathbb{R}^{2x1}.$$

Hence, by assuming that $\mathbf{F}$ is invertible, the terms in equation 25 can be reorganized as

$$\int_{\partial\Omega(s)} \mathbf{D}\nabla V \cdot \mathrm{d}s\boldsymbol{n} = \int_{\partial\Omega_0} J\mathbf{F}^{-1}\mathbf{D}\mathbf{F}^{-T}\nabla V \cdot \mathrm{d}S\boldsymbol{N} \quad (26)$$

Finally, by applying Gauss's divergence theorem again, the divergence term in $\Omega(s)$ and $\Omega_0$ can be expressed as

$$\int_{\Omega(s)} \nabla \cdot (\mathbf{D}\nabla V)\mathrm{d}\Omega = \int_{\Omega_0} \nabla \cdot \left(J\mathbf{F}^{-1}\mathbf{D}\mathbf{F}^{-T}\nabla V\right)\mathrm{d}\Omega_0 \tag{27}$$

By following the same procedure, the boundary condition in equation 19 can be rewritten as

$$\int_{\Omega(s)} (\mathbf{D}\nabla V \cdot \boldsymbol{n})\mathrm{d}\Omega = \int_{\Omega_0} (J\mathbf{F}^{-1}\mathbf{D}\mathbf{F}^{-T}\nabla V \cdot \boldsymbol{N})\mathrm{d}\Omega_0 \tag{28}$$

The remaining parts of equation 19 do not include any divergences or gradients, and are mapped directly through a volume change as defined in equation 21. Consequently, equation 19 can be expressed over $\Omega_0$ as

$$\begin{cases} \frac{\partial}{\partial\tau}(JV) = \nabla \cdot (J\mathbf{F}^{-1}\mathbf{D}\mathbf{F}^{-T}\nabla V) - JkV(V-a)(V-1) - JVW & \text{in } \Omega_0, \\ \frac{\partial}{\partial\tau}(JW) = J\left(\epsilon_0 + \frac{\mu_1 W}{V+\mu_2}\right)(-W - kV(V-a-1)) & \text{in } \Omega_0, \\ J\mathbf{F}^{-1}\mathbf{D}\mathbf{F}^{-T}\nabla V \cdot \boldsymbol{N} = 0 & \text{on } \partial\Omega_0, \end{cases} \tag{29}$$

For time-independent mappings, we finally arrive at

$$\begin{cases} \frac{\partial V}{\partial\tau} = \frac{1}{J}\nabla \cdot (J\mathbf{F}^{-1}\mathbf{D}\mathbf{F}^{-T}\nabla V) - kV(V-a)(V-1) - VW & \text{in } \Omega_0, \\ \frac{\partial W}{\partial\tau} = \left(\epsilon_0 + \frac{\mu_1 W}{V+\mu_2}\right)(-W - kV(V-a-1)) & \text{in } \Omega_0, \\ J\mathbf{F}^{-1}\mathbf{D}\mathbf{F}^{-T}\nabla V \cdot \boldsymbol{N} = 0 & \text{on } \partial\Omega_0, \end{cases} \tag{30}$$

# B  ADDITIONAL DETAILS ON DATASET GENERATION

## B.1  2D GEOMETRIES

Table 5 presents the data ranges used when creating the first three internal ($\mathcal{G}_k$) and external ($\mathcal{G}_k^*$) families in 2D. Additionally, the rotational family ($\mathcal{G}_{rot}$ and $\mathcal{G}_{rot}^*$) was created by defining $\boldsymbol{A}$ as a rotational matrix

$$\mathcal{G}_{rot} \text{ and } \mathcal{G}_{rot}^* : \quad \boldsymbol{A} = \begin{bmatrix} \cos(\theta) & -\sin(\theta) \\ \sin(\theta) & \cos(\theta) \end{bmatrix} \tag{31}$$

with $\theta \in [-\frac{\pi}{2}, \frac{\pi}{2}]$ for $\mathcal{G}_{rot}$ and $\theta \notin [-\frac{\pi}{2}, \frac{\pi}{2}]$ for $\mathcal{G}_{rot}^*$. All values were sampled uniformly from the given ranges.

Table 5: Parameter ranges for the first three internal ($\mathcal{G}_k$) and external ($\mathcal{G}_k^*$) families in 2D. Values were sampled uniformly from the given ranges.

| | $a_1, a_4$ | $a_2, a_3$ | $m_1, m_4$ |
|---|---|---|---|
| $\mathcal{G}_{exp}$ | $[1.0, 1.4]$ | $0.0$ | $0.0$ |
| $\mathcal{G}_{exp}^*$ | $[1.4, 1.8]$ | $0.0$ | $0.0$ |
| $\mathcal{G}_{shear}$ | $1.0$ | $[-0.2, 0.2]$ | $0.0$ |
| $\mathcal{G}_{shear}^*$ | $1.0$ | $[-0.5, -0.2] \cup [0.2, 0.5]$ | $0.0$ |
| $\mathcal{G}_{nonlin}$ | $1.0$ | $0.0$ | $[-0.015, 0.015]$ |
| $\mathcal{G}_{nonlin}^*$ | $1.0$ | $0.0$ | $[-0.025, -0.015] \cup [0.015, 0.025]$ |

## B.2  3D GEOMETRIES

In 3D, we constructed seven families by applying linear affine transformations to a $10 \times 10 \times 10$ mm cube. The linear transformations were defined as

$$\boldsymbol{x} = \boldsymbol{A}\boldsymbol{X} \tag{32}$$

Table 6: Parameter ranges for the internal ($\mathcal{H}_k$) and external ($\mathcal{H}_k^*$) expansion/shearing families in 3D. All values were sampled uniformly from the given ranges.

| | $a_1, a_5, a_9$ | $a_2, a_4$ | $a_3, a_7$ | $a_6, a_8$ |
|---|---|---|---|---|
| $\mathcal{H}_{exp}$ | $[1.0, 1.4]$ | 0.0 | 0.0 | 0.0 |
| $\mathcal{H}_{exp}^*$ | $[1.4, 1.8]$ | 0.0 | 0.0 | 0.0 |
| $\mathcal{H}_{shear}^{xy}$ | 1.0 | $[-0.2, 0.2]$ | 0.0 | 0.0 |
| $\mathcal{H}_{shear}^{xy*}$ | 1.0 | $[-0.5, -0.2] \cup [0.5, 0.2]$ | 0.0 | 0.0 |
| $\mathcal{H}_{shear}^{xz}$ | 1.0 | 0.0 | $[-0.2, 0.2]$ | 0.0 |
| $\mathcal{H}_{shear}^{xz*}$ | 1.0 | 0.0 | $[-0.5, -0.2] \cup [0.5, 0.2]$ | 0.0 |
| $\mathcal{H}_{shear}^{yz}$ | 1.0 | 0.0 | 0.0 | $[-0.2, 0.2]$ |
| $\mathcal{H}_{shear}^{yz*}$ | 1.0 | 0.0 | 0.0 | $[-0.5, -0.2] \cup [0.5, 0.2]$ |

with

$$\boldsymbol{A} = \begin{bmatrix} a_1 & a_2 & a_3 \\ a_4 & a_5 & a_6 \\ a_7 & a_8 & a_9 \end{bmatrix}. \tag{33}$$

Similarly to the 2D scenario, each family was constructed using a distinct deformation type: expansion ($\mathcal{H}_{exp}$), shearing ($\mathcal{H}_{shear}$), and rotation ($\mathcal{H}_{rot}$). The parameter ranges used for the expansion and shearing families are presented in Table 6. Additionally, for the rotational families, $\boldsymbol{A}$ was defined as

$$\mathcal{H}_{rot}^x \text{ and } \mathcal{H}_{rot}^{x*}: \quad \boldsymbol{A} = \begin{bmatrix} 1.0 & 0.0 & 0.0 \\ 0.0 & \cos(\theta) & -\sin(\theta) \\ 0.0 & \sin(\theta) & \cos(\theta) \end{bmatrix},$$

$$\mathcal{H}_{rot}^y \text{ and } \mathcal{H}_{rot}^{y*}: \quad \boldsymbol{A} = \begin{bmatrix} \cos(\theta) & 0.0 & \sin(\theta) \\ 0.0 & 1.0 & 0.0 \\ -\sin(\theta) & 0.0 & \cos(\theta) \end{bmatrix},$$

$$\mathcal{H}_{rot}^z \text{ and } \mathcal{H}_{rot}^{z*}: \quad \boldsymbol{A} = \begin{bmatrix} \cos(\theta) & -\sin(\theta) & 0.0 \\ \sin(\theta) & \cos(\theta) & 0.0 \\ 0.0 & 0.0 & 1.0 \end{bmatrix},$$

with $\theta \in [-\frac{\pi}{2}, \frac{\pi}{2}]$ for internal families and $\theta \notin [-\frac{\pi}{2}, \frac{\pi}{2}]$ for external families. All values were sampled uniformly from the given ranges.

## C  HYPERPARAMETERS AND IMPLEMENTATION DETAILS

Table 7 presents the hyperparameters used in each PINN. The PINNs were implemented with *PyTorch*, and experiments were run on NVIDIA HGX H200 GPUs.

## D  ADDITIONAL DETAILS ON SYNTHETIC DATA GENERATION

We generated the synthetic data using *openCARP* (Plank* et al., 2021; openCARP consortium et al., 2024) with parameters as listed in Table 8 and 9. The diffusion tensor **D** was defined as

$$\mathbf{D} = \begin{bmatrix} \frac{\sigma_{il}\sigma_{el}}{\sigma_{il}+\sigma_{el}} & 0 & 0 \\ 0 & \frac{\sigma_{it}\sigma_{et}}{\sigma_{it}+\sigma_{et}} & 0 \\ 0 & 0 & \frac{\sigma_{in}\sigma_{en}}{\sigma_{in}+\sigma_{en}} \end{bmatrix} \tag{34}$$

in 3D, while in 2D the diffusion tensor was defined as a 2x2 tensor with entries corresponding to longitudinal and transverse directions. Before running the simulation, we meshed the physical geometry using triangular elements in 2D and tetrahedral elements in 3D. The maximum element size was set to 0.05 and 0.4 in 2D and 3D, respectively.

Table 7: Overview of PINN configurations.

|  |  | LPM-PINN | Affine-PINN | Basic-PINN |
|---|---|---|---|---|
| Input dim | 2D | 9 | 9 | 3 |
|  | 3D | 16 | 16 | 4 |
| Hidden layers | isotropic | 10 | 10 | 10 |
|  | anisotropic | 8 | 8 | 8 |
| Hidden dim | isotropic | 25 | 25 | 25 |
|  | anisotropic | 64 | 64 | 64 |
| Output dim |  | 2 | 2 | 2 |
| Epochs |  | 5000 | 5000 | 5000 |
| Batch size |  | 264 | 264 | 264 |
| Optimizer |  | Adam | Adam | Adam |
| Learning rate | $< 100$ epochs | $10^{-3}$ | $10^{-3}$ | $10^{-3}$ |
|  | $> 100$ epochs | $10^{-4}$ | $10^{-4}$ | $10^{-4}$ |
| Activation |  | tanh | tanh | tanh |
| $\mathcal{N}_{data}$ |  | 14 | 14 | 14 |
| $\mathcal{N}_{phys}$ (resampled) |  | 700 | 700 | 700 |
| $\mathcal{N}_{bc}$ (resampled) |  | 80 | 80 | 80 |
| $\mathcal{N}_{ic}$ (resampled) |  | 30 | 30 | 30 |

Table 8: Parameter values used to create synthetic data. PDE parameters were selected in accordance with Aliev & Panfilov, 1996.

| Parameter | Description | Value |
|---|---|---|
| $C_m$ | membrane capacitance | $1\,\mu\mathrm{Fcm}^{-2}$ |
| $\beta$ | surface area to volume ratio | $0.14\,\mu\mathrm{m}^{-1}$ |
| $f_x, f_y, f_z$ | fiber orientation | 1, 0, 0 |
| $\Delta t$ | time resolution | $1\,\mathrm{ms}$ |
| $I_{app}$ | applied stimuli | $5000\,\mu\mathrm{Acm}^{-2}$ for $0.2\,\mathrm{ms}$ (planar wave) |
| $k$ | PDE parameter | 8.0 |
| $a$ | PDE parameter | 0.15 |
| $\varepsilon_0$ | PDE parameter | 0.002 |
| $\mu_1$ | PDE parameter | 0.2 |
| $\mu_2$ | PDE parameter | 0.3 |

Table 9: Conductivities used to create isotropic and anisotropic synthetic data.

| Parameter | Description | Isotropic case | Anisotropic case |
|---|---|---|---|
| $\sigma_{il}$ | intracellular longitudinal conductivity | $0.2\,\mathrm{Sm}^{-1}$ | $0.17\,\mathrm{Sm}^{-1}$ |
| $\sigma_{it}$ | intracellular transversal conductivity | $0.2\,\mathrm{Sm}^{-1}$ | $0.019\,\mathrm{Sm}^{-1}$ |
| $\sigma_{in}$ | intracellular normal conductivity | $0.2\,\mathrm{Sm}^{-1}$ | $0.019\,\mathrm{Sm}^{-1}$ |
| $\sigma_{el}$ | extracellular longitudinal conductivity | $1.0\,\mathrm{Sm}^{-1}$ | $0.62\,\mathrm{Sm}^{-1}$ |
| $\sigma_{et}$ | extracellular transversal conductivity | $1.0\,\mathrm{Sm}^{-1}$ | $0.24\,\mathrm{Sm}^{-1}$ |
| $\sigma_{en}$ | extracellular normal conductivity | $1.0\,\mathrm{Sm}^{-1}$ | $0.24\,\mathrm{Sm}^{-1}$ |

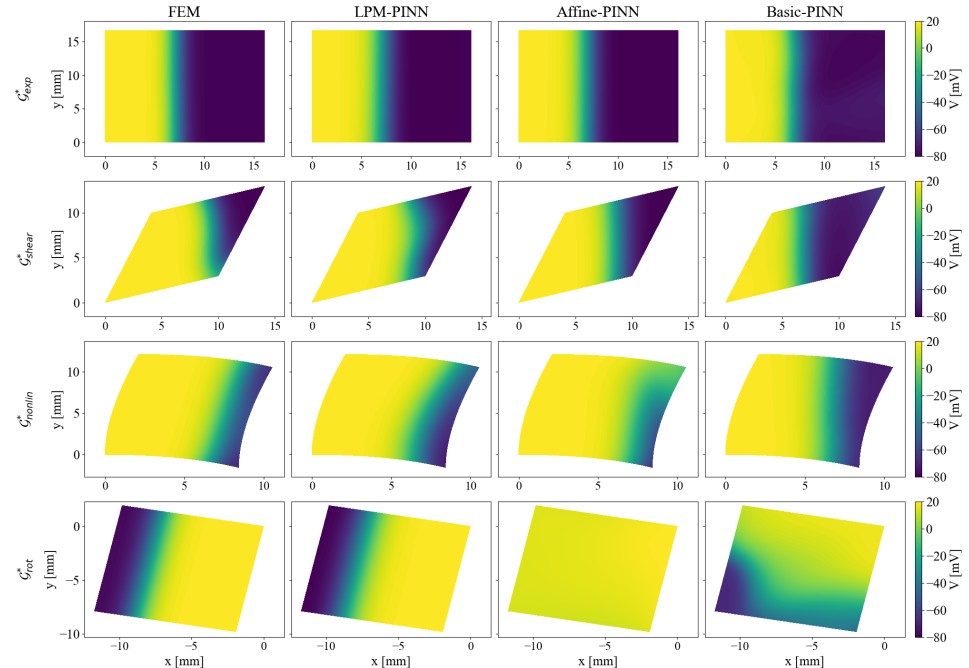

Figure 5: Snapshots of predicted transmembrane voltages ($V$) at $t = 50$ ms. Each row corresponds to a geometry taken from the presented external family ($\mathcal{G}^*_{exp}, \mathcal{G}^*_{shear}, \mathcal{G}^*_{nonlin}, \mathcal{G}^*_{rot}$) in the isotropic scenario. The left column shows the FEM ground truth approximation.

# E  SUPPLEMENTARY RESULTS

## E.1  2D RESULTS

Figure 5 presents snapshots of predicted transmembrane potential ($V$) for selected geometries when the PINNs were trained on single 2D families, while Figure 6 illustrates snapshots when trained on a combination of two families. Both figures represent isotropic PDE dynamics.

### E.1.1  PCA AS GEOMETRIC DESCRIPTOR

Table 10 presents the results obtained on 2D isotropic PDE dynamics when replacing affine parameters with the two PCA modes that capture more than 90% of the geometric variability in each family. A slight increase in error is observed when using PCA modes instead of affine parameters as the geometric descriptor, particularly for the external families. Nonetheless, the overall results indicate that both LPM-PINN and Affine-PINN remain capable of producing accurate predictions when supplied with alternative geometric descriptors. This demonstrates the potential of extending the methods to non-parametric geometries.

## E.2  3D RESULTS

Figure 7 visualizes snapshots of the predicted transmembrane potential ($V$) for selected geometries, while Figures 8 and 9 show the numerical approximation of missing shape gradients at the boundaries for expansion, shearing, and rotational families.

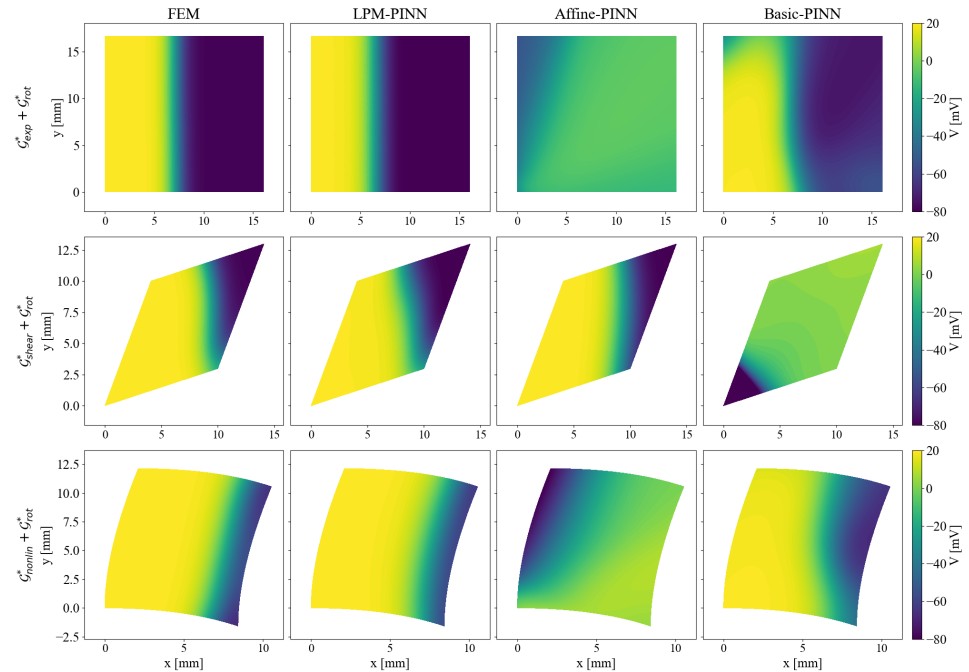

Figure 6: Snapshot of predicted transmembrane voltages ($V$) at $t = 50$ ms for a geometry taken from $\mathcal{G}^*_{exp}$, $\mathcal{G}^*_{shear}$, and $\mathcal{G}^*_{nonlin}$ in the isotropic scenario. The PINNs were trained on a combination of geometries from the given families. The left column shows the FEM ground truth approximation.

Table 10: Mean relative $L_2$ PINN-FEM discrepancy $\pm$ std evaluated over the internal ($\mathcal{G}_k$) and external ($\mathcal{G}^*_k$) test geometries of each geometry family in 2D isotropic scenarios. In this case, the two first PCA modes for the corresponding family were used as additional inputs to describe the geometrical variability.

|  | LPM-PINN | Affine-PINN | Basic-PINN |
|---|---|---|---|
| $\mathcal{G}_{exp}$ | $0.020 \pm 0.003$ | $\mathbf{0.020 \pm 0.003}$ | $0.057 \pm 0.032$ |
| $\mathcal{G}^*_{exp}$ | $0.153 \pm 0.047$ | $\mathbf{0.113 \pm 0.042}$ | $0.196 \pm 0.050$ |
| $\mathcal{G}_{shear}$ | $\mathbf{0.033 \pm 0.009}$ | $0.033 \pm 0.009$ | $0.087 \pm 0.023$ |
| $\mathcal{G}^*_{shear}$ | $0.089 \pm 0.037$ | $\mathbf{0.088 \pm 0.030}$ | $0.204 \pm 0.034$ |
| $\mathcal{G}_{nonlin}$ | $0.023 \pm 0.003$ | $\mathbf{0.023 \pm 0.002}$ | $0.062 \pm 0.023$ |
| $\mathcal{G}^*_{nonlin}$ | $0.139 \pm 0.095$ | $\mathbf{0.138 \pm 0.096}$ | $0.128 \pm 0.041$ |
| $\mathcal{G}_{rot}$ | $\mathbf{0.019 \pm 0.002}$ | $0.040 \pm 0.018$ | $1.876 \pm 0.370$ |
| $\mathcal{G}^*_{rot}$ | $\mathbf{0.026 \pm 0.007}$ | $0.524 \pm 0.128$ | $1.934 \pm 0.252$ |

## F  COMPUTATIONAL DETAILS OF MISSING BOUNDARY INFORMATION

In the following section, we present details on how equation 11 was discretized and numerically approximated. For convenience, we restate the equation here as

$$\frac{\partial \mathcal{L}_{phys}}{\partial s} = \int_{\Omega(s)} \frac{\partial}{\partial s} \mathcal{R}(\boldsymbol{x}, t, u, s)^2 \mathrm{d}\Omega + \int_{\partial\Omega(s)} \mathcal{R}(\boldsymbol{x}, t, u, s)^2 \frac{\partial \boldsymbol{x}}{\partial s} \cdot \boldsymbol{n} \mathrm{d}S \tag{35}$$

and define

$$I(s) \equiv \int_{\Omega(s)} \frac{\partial}{\partial s} \mathcal{R}(\boldsymbol{x}, t, u, s)^2 \mathrm{d}\Omega \tag{36}$$

$$B(s) \equiv \int_{\partial\Omega(s)} \mathcal{R}(\boldsymbol{x}, t, u, s)^2 \frac{\partial \boldsymbol{x}}{\partial s} \cdot \boldsymbol{n} \mathrm{d}S \tag{37}$$

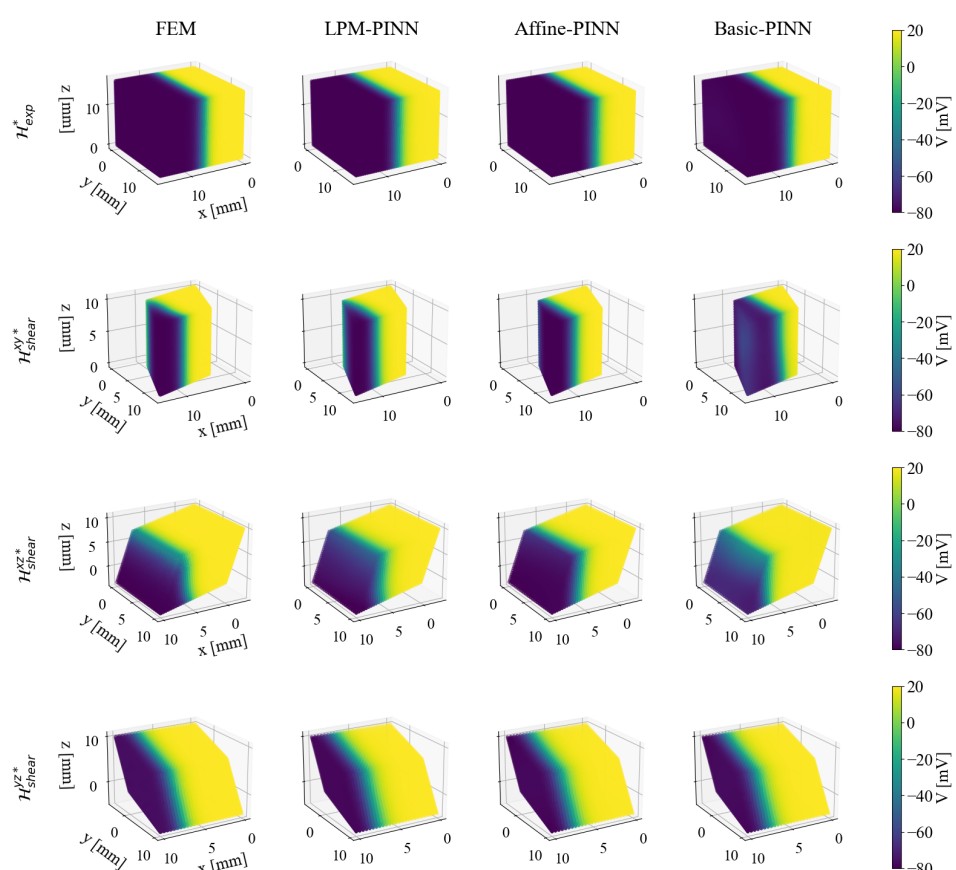

Figure 7: Snapshots of predicted transmembrane voltages ($V$) at $t = 50$ ms. Each row corresponds to a geometry taken from the presented external family ($\mathcal{H}^*_{exp}$, $\mathcal{H}^{xy*}_{shear}$, $\mathcal{H}^{xz*}_{shear}$, $\mathcal{H}^{yz*}_{shear}$) in isotropic scenarios. The left column shows the FEM ground truth approximation.

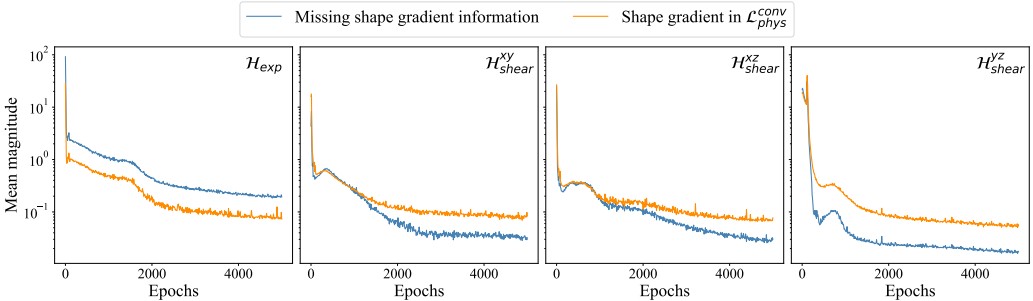

Figure 8: Numerical approximations of the missing shape gradients at the boundary and shape gradients used in $\mathcal{L}^{conv}_{phys}$ when training the Affine-PINN. The figure shows the mean magnitude across training geometries in $\mathcal{H}_{exp}$, $\mathcal{H}^{xy}_{shear}$, $\mathcal{H}^{xz}_{shear}$, and $\mathcal{H}^{yz}_{shear}$.

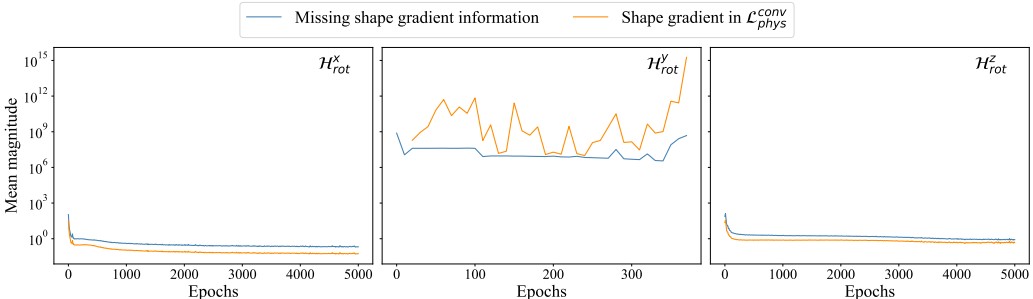

Figure 9: Numerical approximations of the missing shape gradients at the boundary and shape gradients used in $\mathcal{L}_{phys}^{conv}$ when training the Affine-PINN. The figure shows the mean magnitude across training geometries in $\mathcal{H}_{rot}^x$, $\mathcal{H}_{rot}^y$, and $\mathcal{H}_{rot}^z$.

such that

$$\frac{\partial \mathcal{L}_{phys}}{\partial s} = I(s) + B(s). \tag{38}$$

Next, we discretize the terms and make a numerical approximation using Monte Carlo for integrals and central finite differences for derivatives. We assume that the spatial positions are uniformly distributed and normalize with respect to the area/boundary, such that

$$I(s_k) \approx \frac{1}{N_I(s)} \sum_i^{N_I(s)} \sum_j^{\mathcal{T}} \frac{\mathcal{R}(\boldsymbol{x}_i, t_j, u_{ij}, s_k + \Delta s)^2 - \mathcal{R}(\boldsymbol{x}_i, t_j, u_{ij}, s_k - \Delta s)^2}{2\Delta s} \tag{39}$$

and

$$B(s_k) \approx \frac{1}{N_B(s)} \sum_i^{N_B(s)} \sum_j^{\mathcal{T}} \mathcal{R}(\boldsymbol{x}_i, t_j, u_{ij}, s_k)^2 \underbrace{\frac{\boldsymbol{x}_i(s_k + \Delta s) - \boldsymbol{x}_i(s_k - \Delta s)}{2\Delta s} \cdot \boldsymbol{n}_i(s)}_{\text{boundary movement}} \tag{40}$$

where $s_k$ is the $k$-th value in a set of shape parameters given as $s = \{s_1, s_2, ..., s_K\}$. Moreover, $N_I(s)$ and $N_B(s)$ gives the number of spatial positions used to evaluate the two terms and $\mathcal{T}$ is the total number of time steps. Thus, our discretized version for the $k$-th shape value is given as

$$\frac{\Delta \mathcal{L}_{phys}}{\Delta s_k} = I(s_k) + B(s_k) \tag{41}$$

The magnitude of the overall change for the shape parameters ($\frac{\Delta \mathcal{L}_{phys}}{\Delta s}$) was computed by applying the $L_2$ norm to equation 41. Finally, we computed $\frac{\Delta \mathcal{L}_{phys}}{\Delta s}$ for each geometry in a family, and represented the numerical approximation of $I$ and $B$ as the mean across the given geometries. Here, $I$ represents the numerical computation of $\mathcal{L}_{phys}^{conv}$ and $B$ represents the missing boundary information when latent PDE mapping is not applied (see Figures 4, 8, and 9). We used $\Delta s = 10^{-6}$ in all computations.

### F.1 BOUNDARY MOVEMENTS

The magnitude of the numerically approximated boundary movements when making small changes to the shape parameters $s$ for each family in 2D and 3D are presented in Figure 10a and 10b, respectively. The boundary movements were approximated by applying central finite differences, as shown in equation 40, yielding

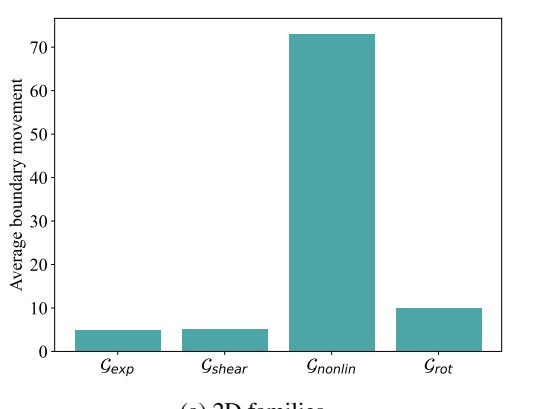
(a) 2D families.

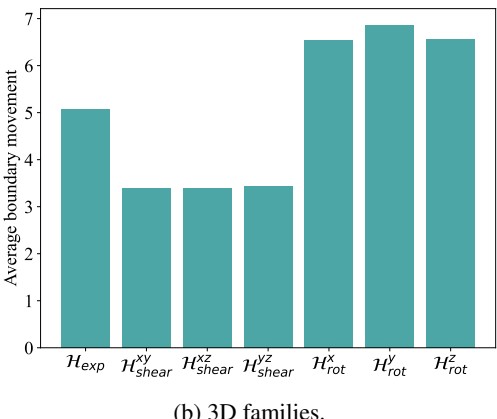
(b) 3D families.

Figure 10: Numerical approximations of boundary movements when making small changes to the shape parameters $s$ for each family in 2D and 3D.

$$\frac{\Delta \boldsymbol{x}}{\Delta s_k} = \frac{1}{N_B(s)} \sum_i^{N_B(s)} \frac{\boldsymbol{x}_i(s_k + \Delta s) - \boldsymbol{x}_i(s_k - \Delta s)}{2\Delta s} \cdot \boldsymbol{n}_i(s) \tag{42}$$

where $N_B(s)$ gives the number of boundary points. Again, the overall magnitude of the boundary movement was computed using the $L_2$ norm and mean across the given geometries with $\Delta s = 10^{-6}$.

## G  COMPUTATIONAL OVERHEAD

Table 11 reports mean per-epoch training times for each PINN across 2D and 3D geometries. A slight increase in computational time per epoch is observed when using LPM-PINN. Furthermore, Table 12 presents mean inference times for geometries in 2D and 3D. The results show that LPM-PINN and Affine-PINN have approximately the same inference times, while Basic-PINN is marginally faster in 2D and substantially faster in 3D. This is likely due to the reduced number of input features for Basic-PINN, which does not require the affine parameters used by LPM-PINN and Affine-PINN in addition to the spatiotemporal inputs. Additionally, the jump in inference time when moving from 2D to 3D is most likely caused by the increased number of spatial locations as well as an increased number of affine parameters in the inputs.

Finally, the additional computational cost associated with computing the deformation gradient and mapping to the reference geometry was estimated to an average time of $4.59 \pm 1.06$ seconds per geometry in 2D and $35.01 \pm 2.33$ seconds per geometry in 3D. As expected, the overhead increases in higher dimensions (3D) and for larger geometries. However, this cost is incurred only once during data preparation. Moreover, the reported times were obtained using a single laptop CPU (Intel Core Ultra 9 185H), indicating that substantial reductions in preprocessing time could be achieved through parallelized CPU execution or by offloading these computations to a GPU.

Table 11: Mean per-epoch training times for 2D and 3D geometries, given as mean $\pm$ std in seconds. The training was performed on a GPU (NVIDIA HGX H200).

|  | LPM-PINN | Affine-PINN | Basic-PINN |
|---|---|---|---|
| 2D geometries | $1.633 \pm 0.634$ | $1.380 \pm 0.489$ | $1.598 \pm 0.649$ |
| 3D geometries | $1.645 \pm 0.042$ | $1.580 \pm 0.047$ | $1.470 \pm 0.076$ |

Table 12: Mean inference times per geometry for 2D and 3D geometries, given as mean $\pm$ std in seconds. A GPU (NVIDIA HGX H200) was used during inference.

|  | LPM-PINN | Affine-PINN | Basic-PINN |
|---|---|---|---|
| 2D geometries | $0.017 \pm 0.009$ | $0.017 \pm 0.009$ | $0.016 \pm 0.009$ |
| 3D geometries | $0.335 \pm 0.047$ | $0.351 \pm 0.055$ | $0.132 \pm 0.029$ |

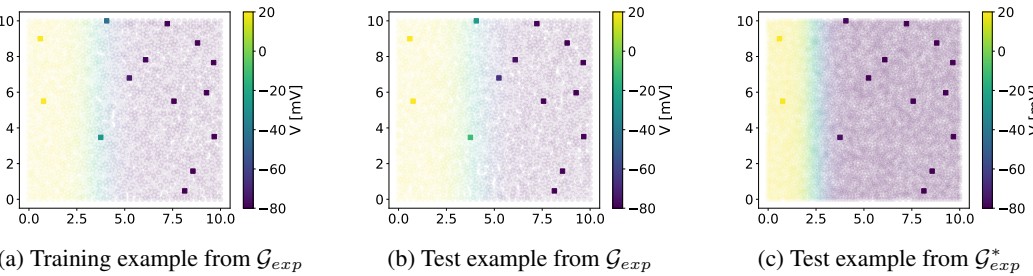

(a) Training example from $\mathcal{G}_{exp}$     (b) Test example from $\mathcal{G}_{exp}$     (c) Test example from $\mathcal{G}_{exp}^*$

Figure 11: Illustration of $V$ at $t = 30$ ms at the fixed sensor points (squares) in the reference geometry used during training and testing of the DeepONet on the $\mathcal{G}_{exp}$ family. $V$ was computed using an interpolation scheme at the fixed sensor points.

## H  DEEPONET EXPERIMENTS

In the following section, we introduce our implementation of a DeepONet as proposed by Lu et al., 2021, as well as the results when running the DeepONet on 2D isotropic experiments using sparse data observations.

### H.1  IMPLEMENTATION DETAILS

We sampled 14 uniformly fixed sensor points in the reference geometry to train the DeepONet, as illustrated in Figure 11. The number of sensor points in the DeepONet was chosen based on the number of supervised data locations ($\mathcal{N}_{data} = 14$) used in the PINNs. Thus, the DeepONet was trained using full time-trajectories of the transmembrane potential $V$ at 14 fixed sensor locations. We employed an interpolation scheme to compute the transmembrane potential $V$ at the fixed sensor locations. It should be noted that the PINNs received $\mathcal{N}_{ic} = 30$ resampled data points at $\tau = 0$ to enforce the initial condition as part of their physics loss during training. These points were not included during training of the DeepONet due to the need for fixed sensor locations.

The DeepONet consisted of a branch network and a trunk network where each network had four hidden layers with 50 neurons in each layer. We gave the full time-trajectories of the transmembrane potential $V$ at the fixed sensor locations as input to the branch network. The trunk network received the spatiotemporal data from the corresponding sensors, and the affine parameters describing the overall physical geometry, as input. We used the tanh as activation function, and Adam as optimizer with a learning rate of 0.001. We trained the DeepONet for 5000 epochs and used the validation data to find the best model obtained during training in a similar manner as for the PINNs. We used the same datasets for training and testing as used for the PINNs. Hence, the DeepONet was trained on the same 10 geometries, validated on the same 5 geometries, and tested on the same 35 geometries in the single family experiments. The same geometries were also used for the combined families as during training and testing of the PINNs.

At inference, the DeepONet received the full time-trajectory of $V$ at the fixed sensor location for each test geometry. Additionally, during inference, the trunk network received spatiotemporal inputs from the entire reference geometry as input. In this way, the DeepONet made predictions of $V$ over the entire geometry, not just at the fixed sensor points, in accordance with the PINNs.

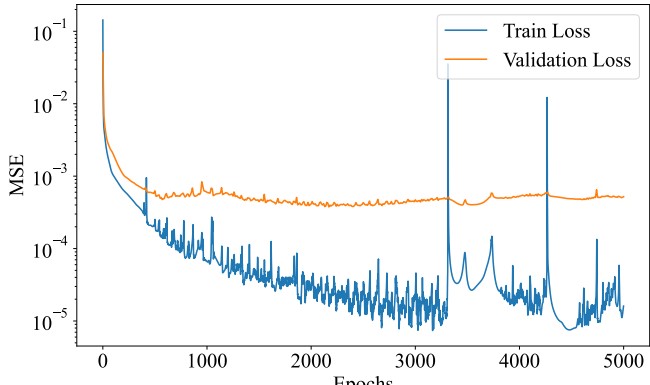

Figure 12: Visualization of training and validation losses during training of the DeepONet on supervised data from $\mathcal{G}_{exp}$. The decrease in training loss (blue) in combination with the stagnant validation loss (orange) indicates that there is not enough observed data available for the DeepONet to generalize.

## H.2 RESULTS

Table 13 and 14 report the mean relative $L_2$ error of the DeepONet predictions when trained on individual and combined 2D geometry families, respectively. The results indicate moderate $L_2$ errors on the internal families and an increase in errors on the external families. This suggests that the Deep-ONet can leverage the fixed sensor locations to make predictions at new spatial positions. Although the $L_2$ errors in Table 13 and 14 may appear acceptable at first glance, Figure 13 shows that the corresponding solutions are not necessarily physically consistent, exhibiting degraded wavefronts. Hence, even though the DeepONet can learn from fixed sensor locations and generalize to new ones, its predictions do not necessarily respect the governing physics, as previously noted by Wang et al., 2021. This limitation underscores the motivation for incorporating physics-informed learning in the first place.

Table 13: Mean relative $L_2$ DeepONet-FEM discrepancy $\pm$ std evaluated over the internal ($\mathcal{G}_k$) and external ($\mathcal{G}_k^*$) test geometries of each geometry family in 2D.

| | DeepONet |
|---|---|
| $\mathcal{G}_{exp}$ | $0.048 \pm 0.007$ |
| $\mathcal{G}_{exp}^*$ | $0.074 \pm 0.013$ |
| $\mathcal{G}_{shear}$ | $0.055 \pm 0.002$ |
| $\mathcal{G}_{shear}^*$ | $0.068 \pm 0.015$ |
| $\mathcal{G}_{nonlin}$ | $0.039 \pm 0.004$ |
| $\mathcal{G}_{nonlin}^*$ | $0.053 \pm 0.010$ |
| $\mathcal{G}_{rot}$ | $0.025 \pm 0.002$ |
| $\mathcal{G}_{rot}^*$ | $0.030 \pm 0.002$ |

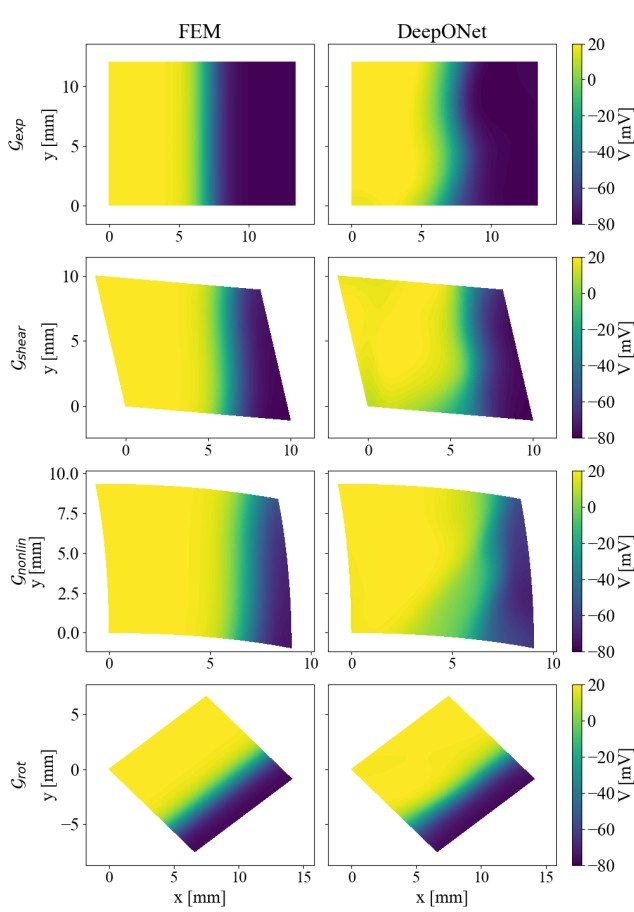

Figure 13: Snapshots of predicted transmembrane voltages $(V)$ at $t = 50$ ms. Each row corresponds to a geometry taken from the presented internal family $(\mathcal{G}_{exp}, \mathcal{G}_{shear}, \mathcal{G}_{nonlin}, \mathcal{G}_{rot})$. The left column shows the FEM ground truth approximation.

Table 14: Mean relative $L_2$ DeepONet-FEM discrepancy $\pm$ std evaluated over the internal ($\mathcal{G}_k$) and external ($\mathcal{G}_k^*$) test geometries from a combination of families in 2D. In this setting, the DeepONet was trained on 20 geometries, validated on 10 geometries, and tested on 70 geometries sampled from the corresponding families.

|  | DeepONet |
| --- | --- |
| $\mathcal{G}_{exp}+\mathcal{G}_{rot}$ | $0.062 \pm 0.005$ |
| $\mathcal{G}_{exp}^*+\mathcal{G}_{rot}^*$ | $0.072 \pm 0.017$ |
| $\mathcal{G}_{shear}+\mathcal{G}_{rot}$ | $0.059 \pm 0.001$ |
| $\mathcal{G}_{shear}^*+\mathcal{G}_{rot}^*$ | $0.063 \pm 0.009$ |
| $\mathcal{G}_{nonlin}+\mathcal{G}_{rot}$ | $0.046 \pm 0.004$ |
| $\mathcal{G}_{nonlin}^*+\mathcal{G}_{rot}^*$ | $0.060 \pm 0.018$ |

