# OpenReview forum: "Latent PDE Mapping for Shape-Generalizable Physics-Informed Neural Networks"
_ICLR.cc/2026/Conference — Submitted to ICLR 2026_

### Official Review · Reviewer_f85A · 2025-10-22

**Soundness:** 3
**Presentation:** 2
**Contribution:** 1
**Rating:** 2
**Confidence:** 5

**Summary:**

A physics-informed neural network training for partial differential equations defined on varying domains.

**Strengths:**

- Easy to read and understand; e.g., input concatenation approach for dealing with parametric dependence.
- An uncommon benchmark problem (Aliev--Panvilov).

**Weaknesses:**

- Pulling back to the reference geometry is standard, not new (e.g., shell-FEM). Once pulled back to the reference geometry, the problem becomes parametric PDEs. Therefore, more appropriate baselines would be methods to solve parametric PDEs, such as DeepONet.
- No data regime? If there is data, then it is necessary to present a useful application.
- Comparison with other PINN methods for varying geometries? Mentioned that other methods are limited to linear and static PDEs, yet that does not mean inapplicability of such methods to the Aliev-Panfilov model.
- Somewhat overselling—the deformation gradient F depends on “t” (eq. (5)), but the benchmark problem considers static (in time) geometries.
- No thorough investigation of why the method generalizes to out-of-range cases.

**Questions:**

- How did you sample collocation points for PDE residuals?
- line 259: What does it mean by “tanh for handling second-order derivatives?”
- For nonlinear geometry, F depends on x. How did you compute PDE residuals with AD in this case?
- What are the values for D, mu_1, mu_2, epsilon_0, k, and a? Can’t find in Table 7.
- Why are snapshot times varying? Difficult to compare. Fix them for the final time.
- How are FEM solutions stored? In the reference geometry?
- More explanation is necessary for the two geometry cases: G_exp + G_rot.
- In Table 1, why did Affine-PINN perform well?
- Given that Affine-PINN performed well in Table 1, Figure 4 may indicate that missing shape gradient information is not so informative?

---

> ### Author Response · Authors · 2025-11-22
>
> We would like to thank the reviewer for the thorough and detailed assessment of our paper. We appreciate the constructive feedback and address the identified weaknesses and questions below.
>
> &nbsp;
>
> #### **Weaknesses:**
>
> **[W1] Pulling back to the reference geometry is standard, not new (e.g., shell-FEM). Once pulled back to the reference geometry, the problem becomes parametric PDEs. Therefore, more appropriate baselines would be methods to solve parametric PDEs, such as DeepONet.**
>
> We agree that the pull-back to a reference geometry is not new in the context of numerical PDE solvers. Our claim is not that the mathematical idea is new, but that this technique has not been leveraged within PINN training. Prior work has employed limited coordinate transformations, but a full pull-back of the PDE through the deformation gradient during training, as proposed here, has been largely unexplored.
>
> We also agree that the pull-back effectively casts the problem as a parametric PDE. However, our target application lies in a sparse-data regime, where only tens of examples are available. Neural operator approaches such as DeepONet typically rely on substantially larger datasets and perform poorly when data are scarce. To clarify this point, we have (i) more explicitly framed our work as addressing the sparse-data setting in the related work (Section 2), and (ii) included a DeepONet implementation on the 2D isotropic experiments in Appendix H, demonstrating that standard operator-learning methods fail to learn a meaningful representation under the same data limitations (see Table 13, Table 14 and Figure 12).
>
> **[W2] No data regime? If there is data, then it is necessary to present a useful application.**
>
> We kindly ask the reviewer to clarify this comment, as we may have misunderstood the concern.
>
> Our work indeed uses limited observational data to demonstrate the applicability to a realistic cardiac electrophysiology scenario. Many personalization tasks in cardiology operate with only sparse measurements, which motivates the need for physics-informed methods capable of functioning in low-data settings.
>
> **[W3] Comparison with other PINN methods for varying geometries? Mentioned that other methods are limited to linear and static PDEs, yet that does not mean inapplicability of such methods to the Aliev-Panfilov model.**
>
> We included, as a baseline, PINNs augmented with affine geometric parameters, which is consistent with prior approaches for geometry-dependent PINNs. However, the prior approaches based on linear and static PDE did not make use of the deformation gradient. While the prior approaches could be applied directly to Aliev-Panfilov, the results of Figures 2 and 3 indicate that the results would be suboptimal due to missing gradient information. Indeed, the coordinate mappings for a linear PDE do not require any special considerations and are quite straightforward, which is not the case for nonlinear PDE such as the Aliev-Panfilov equations. In our work we provide a simple mathematical framework that can be readily applied to improve the performance of previous PINN methods when applied to nonlinear PDE defined over varying geometries.
>
> **[W4] Somewhat overselling—the deformation gradient F depends on “t” (eq. (5)), but the benchmark problem considers static (in time) geometries.**
>
> We appreciate this observation. Our intention was to present the framework in its most general form, applicable also to time-dependent deformations. Although the deformation gradient in Eq. (5) includes t, our experiments do not involve time-dependent deformations. To avoid confusion, we have clarified in the revised version that the geometries in our benchmarks are static in time (line 224), and that the time dependence in the formulation is included only to illustrate general applicability (line 131).
>
> **[W5] No thorough investigation of why the method generalizes to out-of-range cases.**
>
> Our results in Figure 4 and theoretical analysis in Section 3.2 suggest that the improved out-of-range generalization arises from the LPM formulation’s ability to preserve shape gradients accurately across different geometries. By expressing the PDE on the reference geometry, the method provides the PINN with more stable and consistent training gradients, leading to a better representation of the governing physics. This improved physical fidelity contributes to better extrapolation when encountering unseen or our-of-range geometries.

---

> ### Author Response · Authors · 2025-11-22
>
> In the following, we address the questions raised by the reviewer
>
> &nbsp;
>
> #### **Questions:**
>
> **[Q1] How did you sample collocation points for PDE residuals?**
>
> We sampled 700 collocation points uniformly at random within each geometry using np.random.choice. At every training epoch, we resampled points using the same procedure but with a different random seed.
>
> **[Q2] line 259: What does it mean by “tanh for handling second-order derivatives?”**
>
> We appreciate that the reviewer pointed out the lack of clarity regarding the use of second-order derivatives and the tanh function.  Since the PDE residual involves second-order derivatives, automatic differentiation computes second-order derivatives of the network’s activation functions. To avoid these derivatives becoming zero (as would happen for piecewise linear activations such as ReLU), a smooth nonlinear activation function such as tanh is required so that the second-order derivatives are non-zero and informative during training.
>
> To clarify this issue, we have updated the text with specific equation numbers:
>
> *“Furthermore, we employed the tanh as activation function in all cases to handle
> second-order derivatives (equation 8) needed to calculate the physics loss (equation 9).”*
>
> **[Q3] For nonlinear geometry, F depends on x. How did you compute PDE residuals with AD in this case?**
>
> We thank the reviewer for bringing up this interesting issue. All deformation gradients F are currently pre-computed locally at each spatial location. Thus, no AD is currently needed to compute F. The PDE residuals are evaluated using automatic differentiation as usual, with F treated as a pointwise known quantity.
>
> **[Q4] What are the values for D, mu_1, mu_2, epsilon_0, k, and a? Can’t find in Table 7.**
>
> We thank the reviewer for bringing this missing information to our attention. The diffusion tensor D is defined as
>
> $$
>  D = [[(σ_{il} σ_{el})/(σ_{il} +σ_{el} ), 0], [0, (σ_{it} σ_{et})/(σ_{it} + σ_{et} )]]
> $$
>
> in the 2D case. We have now provided the values for $σ_{il}, σ_{it}, σ_{el}, σ_{et}, σ_{in},  σ_{en}$ in Table 9 of the revised version and added defined D in eq. (34).  Furthermore, mu_1 = 0.2, mu_2 = 0.3, epsilon_0 = 0.002, k = 8.0, and a = 0.15. These values were also missing in the paper and have been added to Table 8 of the revised version. We thank the reviewer for catching this omission.
>
> **[Q5] Why are snapshot times varying? Difficult to compare. Fix them for the final time.**
>
> We agree that the final time is a natural and objective choice to use for PDE visualisation. However, in our simulations, the activation wave has passed through the geometry at the final time and hence we chose a time where the wave dynamics were most clearly visible, and the PINN predictions diverged most clearly. We agree that consistent snapshot times improve comparability. As a result, we have updated all figures to use the same snapshot time (50 ms) in the revised manuscript for easier comparison.
>
>
> **Response will continue in next comment due to character limit.**

---

> > ### Author Response · Authors · 2025-11-22
> >
> > **[Q6] How are FEM solutions stored? In the reference geometry?**
> >
> > FEM solutions are computed and stored on the physical geometries. During training and evaluation of the LPM-PINN, we map FEM solutions between the physical and reference geometries using the coordinate transformation x --> X(x) and its inverse.
> >
> > **[Q7] More explanation is necessary for the two geometry cases: G_exp + G_rot.**
> >
> > We thank the reviewer for pointing out this spot where more clarity is needed.
> > To construct the combined families (e.g. G_exp + G_rot), we merged the geometries from both families, yielding 100 total geometries (50 from each family). Training, validation, and test sets were drawn proportionally:
> >
> > - Training: 20 geometries (10 from each family)
> > - Validation: 10 geometries (5 from each family)
> > - Test: 70 geometries (35 from each family)
> >
> > We similarly combined the corresponding external families to form the external test set (70 geometries in total, 35 from each family). In order to clarify this for our readers, we have added additional descriptions to Section 5 of the revised manuscript.
> >
> > **[Q8] In Table 1, why did Affine-PINN perform well?**
> >
> > The strong performance of the Affine-PINN in the 2D experiments likely indicates that the model successfully learned an internal mapping, and that the geometric variation within each individual family was not large enough for the missing boundary-term information to significantly degrade performance.
> >
> > This limitation becomes clearer in the combined-family experiments (Table 2), where shape variation is substantially larger. In this scenario the Affine-PINN struggles, suggesting that without explicit mapping the model cannot capture the larger boundary changes between families.
> >
> > **[Q9] Given that Affine-PINN performed well in Table 1, Figure 4 may indicate that missing shape gradient information is not so informative?**
> >
> > As noted in our response to Q8, the impact of missing shape-gradient information depends on the magnitude of geometric variability. When geometric variation is small (as in the single-family case), the Affine-PINN can still learn adequately. When variation increases (as in the combined families), the missing shape-gradient information becomes critical, and performance degrades.

---

> ### Comment · Reviewer_f85A · 2025-11-24
>
> Thank you very much for answering my questions.
>
> W2. I am sorry for asking an elusive question. As far as I know, previous works in PINNs are either (1) focus on forward problems (no data) or (2) include inverse problems where traditional methods are costly. The reason is that people know that (1) is difficult. However, this work included data in the training process. In this case, I think showcasing a useful application is necessary; otherwise, it might be less interesting.
>
> Q9. In my understanding, a large magnitude of missing shape gradient information must indicate inaccurate Affine and Vanilla-PINNs. However, $\mathcal{G}\_{nonlin}$ has a large magnitude (Figure 4), but Affine-PINN performed well (Table 1). By contrast, $\mathcal{G}\_{rot}$ has a smaller magnitude in Figure 4 (at the final epoch), but Affine-PINN did not do so well (Table 1). I think the reader may be confused, like me; therefore, I would suggest a more detailed explanation about how to interpret the result.

---

> > ### Author Response · Authors · 2025-11-28
> >
> > We appreciate that the reviewer engages in the discussion and has given us the chance to clarify some additional points.
> >
> > **[W2] I am sorry for asking an elusive question. As far as I know, previous works in PINNs are either (1) focus on forward problems (no data) or (2) include inverse problems where traditional methods are costly. The reason is that people know that (1) is difficult. However, this work included data in the training process. In this case, I think showcasing a useful application is necessary; otherwise, it might be less interesting.**
> >
> > We thank the reviewer for clarifying this comment. The reviewer is correct that physics-informed neural networks (PINNs) have indeed also been widely used for both forward PDE problems (without data) and inverse PDE problems (estimating parameters with abundant data). However, an original motivation behind physics-informed machine learning, as discussed by Karniadakis et al. [1], was to enable learning in regimes where data is limited (e.g., sparse measurements or only initial condition data available) or of low quality (noisy, sparse, or containing outliers). In such cases, incorporating physics into the learning process can significantly improve model performance, especially when traditional data-driven methods struggle. This scenario is common in many real-world physical systems where sensor measurements are limited or noisy [1], such as solid mechanics [2, 7], reconstruction of wavefields [3, 8], and fluid dynamics [4, 5, 6].
> >
> > In our work, we focus on a clinical application (cardiac electrophysiology) where data is inherently limited, as highlighted in the Related Work section:
> >
> > *"This motivates the pursuit of data-efficient approaches capable of learning from fewer geometric samples, an essential consideration in domains where data collection is costly or ethically constrained, such as medicine."*
> >
> > Indeed, in real clinical settings, it is impossible to measure electrical activity across the entire cardiac tissue, since measurements are only available at sparse sensor locations [9]. Physics-informed learning has the potential to make meaningful inferences from these limited measurements, demonstrating the practical utility of PINNs in such contexts.
> >
> >
> > [1] Karniadakis, George Em, et al. "Physics-informed machine learning." Nature Reviews Physics 3.6 (2021): 422-440.
> >
> > [2]. Go, Myeong-Seok, Hong-Kyun Noh, and Jae Hyuk Lim. "Real-time full-field inference of displacement and stress from sparse local measurements using physics-informed neural networks." Mechanical Systems and Signal Processing 224 (2025): 112009.
> >
> > [3]. Xu, Bin, et al. "Sparse wavefield reconstruction based on Physics-Informed neural networks." Ultrasonics 149 (2025): 107582.
> >
> > [4]. Arzani, Amirhossein, Jian-Xun Wang, and Roshan M. D'Souza. "Uncovering near-wall blood flow from sparse data with physics-informed neural networks." Physics of Fluids 33.7 (2021).
> >
> > [5]. Chaurasia, Nagendra Kumar, and Shubhankar Chakraborty. "Reconstruction of the turbulent flow field with sparse measurements using physics-informed neural network." Physics of Fluids 36.8 (2024).
> >
> > [6]. Hosseini, Mohammad Yasin, and Yousef Shiri. "Flow field reconstruction from sparse sensor measurements with physics-informed neural networks." Physics of Fluids 36.7 (2024).
> >
> > [7]. Noh, Hong-Kyun, Myeong-Seok Go, and Jae Hyuk Lim. "Real-time monitoring of thermoelastic deformation of a silicon wafer with sparse measurements in the photolithography process using a physics-informed neural network and Fourier neural operator." Engineering Applications of Artificial Intelligence 152 (2025): 110767.
> >
> > [8] Zargar, Sakib Ashraf, and Fuh-Gwo Yuan. "Physics-informed deep learning for scattered full wavefield reconstruction from a sparse set of sensor data for impact diagnosis in structural health monitoring." Structural Health Monitoring 23.5 (2024): 2963-2979.
> >
> > [9] Herrero Martin, Clara, et al. "EP-PINNs: Cardiac electrophysiology characterisation using physics-informed neural networks." Frontiers in Cardiovascular Medicine 8 (2022): 768419.

---

> > > ### Author Response · Authors · 2025-11-28
> > >
> > > **[Q9] In my understanding, a large magnitude of missing shape gradient information must indicate inaccurate Affine and Vanilla-PINNs. However, G_nonlin  has a large magnitude (Figure 4), but Affine-PINN performed well (Table 1). By contrast,  G_rot has a smaller magnitude in Figure 4 (at the final epoch), but Affine-PINN did not do so well (Table 1). I think the reader may be confused, like me; therefore, I would suggest a more detailed explanation about how to interpret the result.**
> > >
> > > We thank the reviewer for raising this insightful point. The magnitude of the missing shape gradient shown in Fig. 4 provides an estimate of the raw missing gradient component for each shape family. However, how the boundary gradients influence the performance is unclear. In general, this is a difficult issue for neural networks. For example, gradient descent methods might end up “bouncing” back and forth, which results in the cancellation of large gradient components.
> > >
> > > In response to the reviewer’s comment, we have updated the discussion (Section 7) to clarify this issue for the reader:
> > >
> > > *“A central insight emerging from this study concerns the role of missing boundary shape gradients. Adding the boundary gradient via latent PDE mapping can boost the ability of PINNs to generalize to new shapes (see LPM-PINN versus Affine-PINN in Table 1-4). Indeed, in the absence of latent PDE mapping, the omitted boundary terms can be larger than the remaining gradients (Figure 4). However, boundary gradient size does not necessarily translate directly into performance improvement. Thus, there is a need for more research to further investigate this issue.”*

---

### Official Review · Reviewer_C2RH · 2025-10-28

**Soundness:** 2
**Presentation:** 3
**Contribution:** 1
**Rating:** 2
**Confidence:** 5

**Summary:**

The paper proposes latent PDE mapping for PINNs to solve PDEs of varying geometries. The method maps parametric geometries to a common latent geometry, and derives a general physics loss formula to train PINN. Experiment results on 2D and 3D Aliev-Panfilov equations are presented.

**Strengths:**

1. The paper is clearly written and easy to follow.

**Weaknesses:**

1. The contribution of the paper appears weak. It completely omits closely related works in the literature, which already address the same problem. Existing work such as [1] has demonstrated the possibility of mapping arbitrary nonparametric geometries to certain latent space, to learn large scale complex 3D PDEs. In comparison, the proposed method can only handle simple parametric cases.

2. The experiments are weak in terms of both complexity and baselines. Only rectangular domains and their affine transformations are considered. Moreover, only naive PINN and affine PINN are compared with, missing state of the art methods such as [1-3].

3. While the author did not mention, I believe the method only applies to domains that are diffeomorphic to the latent domain, as otherwise the gradient information can not be mapped. This will heavily limit the applicability of the method.

4. As a side note, I don’t think using inverted allocation of train-val-test split can demonstrate the model performance in data-limit scenarios. Few shot generalization ability only depends on the sufficiency of training data, not on how all data are partitioned.


[1] Li, Zongyi, et al. "Geometry-informed neural operator for large-scale 3d pdes." Advances in Neural Information Processing Systems 36 (2023): 35836-35854.

[2] Li, Zongyi, et al. "Fourier neural operator with learned deformations for pdes on general geometries." Journal of Machine Learning Research 24.388 (2023): 1-26.

[3] Li, Zijie, Anthony Zhou, and Amir Barati Farimani. "Generative Latent Neural PDE Solver using Flow Matching." arXiv preprint arXiv:2503.22600 (2025).

**Questions:**

1. How does the proposed method compare with state of the art methods such as [1-3]?

2. How does the proposed method work under more challenging benchmark problems such as the ones considered in [1-3]?

---

> ### Author Response · Authors · 2025-11-22
>
> We thank the reviewer for the constructive feedback on our paper. Below we address the raised weaknesses and questions:
>
> &nbsp;
>
> #### **Weaknesses:**
>
> **[W1] The contribution of the paper appears weak. It completely omits closely related works in the literature, which already address the same problem. Existing work such as [1] has demonstrated the possibility of mapping arbitrary nonparametric geometries to certain latent space, to learn large scale complex 3D PDEs. In comparison, the proposed method can only handle simple parametric cases.**
>
> We appreciate the reviewer raising this point. Our work intentionally focuses on a sparse-data regime, where only tens of geometries are available. In such cases, purely data-driven neural operator approaches (including the methods cited by the reviewer) have been shown to require large and diverse datasets to learn robust operators. This scale of data is not accessible in the targeted clinical setting (e.g., electrophysiology), making physics-based constraints essential. We acknowledge that this important point may not have been entirely clear in the original manuscript. We have therefore included a new paragraph in the “related work” section (Section 2) reviewing recent studies involving operator networks and graph network approaches.
>
> Although study [1] mentioned by the reviewer indeed maps arbitrary nonparametric geometries to a latent space, this approach is entirely data-driven and does not include a corresponding mapping of the PDE itself, which is a key requirement for physics-informed methods operating under data scarcity. In contrast, our contribution lies in demonstrating that mapping geometries and their governing PDEs to a common latent geometry enables accurate learning when data are extremely limited.
>
> While the idea of mapping geometries to a latent space is not new, the full pull-back of the PDE via the deformation gradient inside the PINN training loop is novel within the PINN literature. Our goal is not to compete with neural operators in high-data settings, but to show that latent PDE mapping can substantially enhance physics-informed learning in low-data regimes across varying geometries, a contribution which is complementary to the recent literature. Indeed, the latent PDE mapping approach is agnostic to network architecture, and can also be applied to neural operators and graph networks, as long as they use a physics-based loss function and diffeomorphic domain mappings. To make this point clearer to the reader, we have added an additional comment to the introductions (Section 1)
>
> *“The proposed approach offers a broadly applicable strategy for extending PINNs to problems involving geometries with variable shapes, complementary to the current state of the art approaches involving operator and graph-neural architecture”*
>
> **[W2] The experiments are weak in terms of both complexity and baselines. Only rectangular domains and their affine transformations are considered. Moreover, only naive PINN and affine PINN are compared with, missing state of the art methods such as [1-3].**
>
> We agree that the experiments use simplified geometries to illustrate the LPM technique in a controlled and interpretable setting. To strengthen the experimental section, we have included an additional anisotropic 2D experiment in Section 6 (Table 3 and Figure 2)  that demonstrates the method’s applicability to more complex PDE dynamics.
>
> Regarding baselines, our study specifically targets physics-informed learning under sparse data. Neural operator methods such as [1–3] generally rely on large datasets and would not constitute fair or informative comparisons in this regime. Instead, we compare against the most relevant PINN methods that, like ours, must operate with very limited data and leverage governing physics for training.
>
> Nonetheless, we have updated our “related work” section (Section 2) to clarify our position in the data-limited regime. Furthermore, we have added DeepONet results for the 2D isotropic experiments to ensure that we are indeed operating in the limited-data regime. The results (Table 13 and 14) show that the DeepONet struggles to learn accurate solutions when the observed data is very limited.
>
> **The response will continue in next comment due to character limit.**

---

> ### Author Response · Authors · 2025-11-22
>
> **[W3] While the author did not mention, I believe the method only applies to domains that are diffeomorphic to the latent domain, as otherwise the gradient information can not be mapped. This will heavily limit the applicability of the method.**
>
> We agree with the reviewer’s observation. The method indeed requires the geometries to be diffeomorphic to ensure that gradient information can be consistently mapped. However, recent work by Yin et al. [1] shows the potential and broad applicability of diffeomorphic domains.
>
> [1] Yin M, Charon N, Brody R, Lu L, Trayanova N, Maggioni M. A scalable framework for learning the geometry-dependent solution operators of partial differential equations. Nature computational science. 2024 Dec;4(12):928-40.
>
> **[W4] As a side note, I don’t think using inverted allocation of train-val-test split can demonstrate the model performance in data-limit scenarios. Few shot generalization ability only depends on the sufficiency of training data, not on how all data are partitioned.**
>
> We thank the reviewer for this comment. Our intention with the inverted split was not to imply that the partitioning itself controls generalization ability, but rather to simulate a real-world scenario in which only a small subset of geometries is available for training. We appreciate the opportunity to clarify this.
>
> &nbsp;
>
> #### **Questions:**
>
> **[Q1] How does the proposed method compare with state of the art methods such as [1-3]?**
>
> As discussed in our response to the weaknesses [W2], we believe that direct comparison to neural operator methods is not meaningful in our setting due to the severely limited data regime. Neural operators typically require large and diverse datasets to learn effective operators, whereas our problem setting provides only tens of geometries. Under such conditions, these methods generally struggle to learn meaningful representations.
>
> Nonetheless, to address the reviewer’s concern and provide a concrete point of reference, we have included a DeepONet implementation in Appendix H of the revised manuscript. The results (Table 13, Table 14 and Figure 12) verifies that we are indeed operating in a regime where data-driven operator-learning approaches do not perform well, further motivating the need for a physics-informed method such as LPM-PINN.
>
> Additionally, we have updated our “related work” section (Section 2) to clarify our position within the data-limited regime for the reader and why physics-informed incorporation is needed.
>
> **[Q2] How does the proposed method work under more challenging benchmark problems such as the ones considered in [1-3]?**
>
> We appreciate the reviewer’s question. A thorough investigation of performance on the complex benchmark problems used in [1–3] is beyond the scope of the present paper. Our focus is specifically on demonstrating the utility of latent PDE mapping in sparse-data, physics-informed scenarios, rather than competing with neural operators on large-scale data-driven benchmarks.

---

### Official Review · Reviewer_DTkg · 2025-11-01

**Soundness:** 3
**Presentation:** 2
**Contribution:** 2
**Rating:** 4
**Confidence:** 3

**Summary:**

This paper introduces latent PDE mapping to enhance the generalization of PINNs in different geometries. It maps geometry-specific PDEs to a shared latent space using the deformation gradient and affine shape parameterization. The method is tested on Aliev-Panfilov model of cardiac electrophysiology in 2D and 3D and shows improvement.

**Strengths:**

- It is an important problem to generalize PINNs across geometries without retraining. The use of a simple MSE loss on collocation points neglects the gradient component from the movement of the domain boundary. The paper provides both theoretical and empirical evidence that this is suboptimal.
- The focus on cardiac electrophysiology adds practical relevance.

**Weaknesses:**

- The geometric representation through affine transformations is limited. The authors acknowledge this limitation, but it remains unclear whether latent PDE mapping can be extended to a more general setting.
- The PDE dynamics tested are simplified for the stated application domain. The paper only considers isotropic diffusion.
- Lack of computational efficiency analysis. The method requires computing the deformation gradients but he paper provides no timing comparisons.
- The experimental comparisons are incomplete, particularly recent geometry-aware neural operators, for example GNO and DeepONet. The baselines (vanilla PINNs and APINN) are not adequate.

**Questions:**

Mentioned in Weaknesses.

---

> ### Author Response · Authors · 2025-11-22
>
> We would like to thank the reviewer for the detailed and constructive feedback. We address the noted weaknesses below:
>
> &nbsp;
>
> #### **Weaknesses**
>
> **[W1] The geometric representation through affine transformations is limited. The authors acknowledge this limitation, but it remains unclear whether latent PDE mapping can be extended to a more general setting.**
>
> As noted, affine parameters do not fully capture the richness of more complex geometric variability. However, the latent PDE mapping technique itself is not restricted to affine transformations. The same mathematical formulation applies to more general mappings, including diffeomorphic transformations. Recent work (e.g., [1] Yin et al.) demonstrates how parameterized domains can be extended to PCA-based and fully diffeomorphic geometries, indicating a natural path for broader applicability. To further validate this, we have included an additional experiment in Appendix E.1.1 (Table 10) of the revised manuscript. These results show that our method can successfully learn meaningful representations when provided with more general, PCA-based geometric descriptors.
>
> [1] Yin M, Charon N, Brody R, Lu L, Trayanova N, Maggioni M. A scalable framework for learning the geometry-dependent solution operators of partial differential equations. Nature computational science. 2024 Dec;4(12):928-40.
>
> **[W2] The PDE dynamics tested are simplified for the stated application domain. The paper only considers isotropic diffusion.**
>
> We agree that the PDE dynamics in the current version are simplified to isolate and clearly demonstrate the benefits of the latent PDE mapping technique in a controlled setting. However, in response to the reviewer’s suggestion, we have included an additional anisotropic 2D experiment in Section 6 (Table  3 and Figure 2) of the revised paper. The result demonstrate the method’s ability to handle more complex, clinically relevant PDE dynamics.
>
> **[W3] Lack of computational efficiency analysis. The method requires computing the deformation gradients but he paper provides no timing comparisons.**
>
> We appreciate this comment and agree that computational considerations are important for assessing practical applicability. In the revised paper we have added Appendix G which includes:
>
> - Training times (Table 11 in Appendix G)
> - Inference times (Table 12 in Appendix G)
> - Estimated computational cost of deformation-gradient evaluations and mappings (Appendix G)
>
> Our results (Table 11 and 12) show that the LPM-PINN and Affine-PINN have roughly the same training and inference times. However, the LPM-PINN comes at an additional one-time cost for computing the deformation gradient and mapping. In 2D, this additional cost is a modest 4.59 $\pm$ 1.06 seconds per geometry, while in 3D it is 35.01 $\pm$ 2.33 seconds per geometry. Note that these estimated times was computed on a single laptop CPU, hence the computational overhead could be further reduced by using a GPU. We have included an additional paragraph in the discussion (Section 7) to highlight these computational specifications.
>
> **[W4] The experimental comparisons are incomplete, particularly recent geometry-aware neural operators, for example GNO and DeepONet. The baselines (vanilla PINNs and APINN) are not adequate.**
>
> We appreciate the reviewer highlighting this point. While indeed performance of cited methods such as GNO and DeepONet are favourable in a number of settings, these methods require a wealth of data (in this case, diverse and numerous geometries) to perform well. In the specific challenge setting which our work addresses, this wealth of data is not available, and is often limited to tens of examples.
>
> To address the reviewer’s concern, we have included DeepONet results for isotropic 2D experiments in Appendix H of the revised manuscript (Table 13 and Table 14). In addition, Figure 12 now visualizes the training and validation losses when the DeepONet was trained on the Gexp family. We interpret the figure as a typical overfitting scenario where the model struggles to generalize to new unseen data. These additional experiments help illustrate the limitations of purely data-driven operator learning in this regime and further motivate the need for a physics-informed approach such as LPM-PINN when data availability is extremely limited.

---

### Official Review · Reviewer_k92c · 2025-11-11

**Soundness:** 4
**Presentation:** 4
**Contribution:** 4
**Rating:** 6
**Confidence:** 4

**Summary:**

The paper proposes latent PDE mapping (LPM): it maps geometry-specific PDEs onto a shared latent PDE defined on a reference domain via the deformation gradient, and embeds this into a PINN (LPM-PINN). By moving shape dependence from the geometry into the PDE coefficients, the method aims to capture geometric variability while preserving the underlying physics and enabling correct shape gradients. They validate LPM-PINN on synthetic cardiac electrophysiology (Aliev–Panfilov) in 2D and 3D over rich families of deformed domains.

**Strengths:**

1.  Using deformation-gradient–based PDE mapping directly inside a PINN to create a shared latent PDE for varying geometries is conceptually new.
2. Performance is good compared with Affine-PINN and naive PINNs.
3. Key ideas (latent PDE, deformation gradient, missing boundary gradients) are explained at a level that a PINN/PDE reader can follow.

**Weaknesses:**

1. Problem scope is narrow:
All results are on synthetic cardiac PDE data with parameterized (mostly affine) deformations and simplified Aliev–Panfilov dynamics.

2. Limited ablation on: the necessity of the full deformation-gradient-based mapping vs simpler coordinate transforms, sensitivity to the choice of latent domain, computational overhead trade-offs.

**Questions:**

None

---

> ### Author Response · Authors · 2025-11-22
>
> We thank the reviewer for the thoughtful assessment of our paper. We appreciate that the reviewer finds the paper conceptually sound, well-presented, and the contributions valuable. Below we address the two perceived weaknesses raised by the reviewer:
>
> &nbsp;
>
> #### **Weaknesses**
> **[W1] Problem scope is narrow: All results are on synthetic cardiac PDE data with parameterized (mostly affine) deformations and simplified Aliev–Panfilov dynamics.**
>
> We acknowledge that our current experiments focus on Aliev–Panfilov dynamics on parameterized domains. We humbly propose that this perceived weakness is also a strength. Our intention was to isolate and clearly demonstrate the contribution of the latent PDE mapping technique without confounding effects from noise or complex anatomical geometries. The Aliev–Panfilov model, while simple in form, exhibits stiff, nonlinear, and fast-propagating wavefronts that remain challenging for PINNs and thus provide a meaningful test bed. Using synthetic data also ensures that differences in performance arise from the method itself rather than uncontrolled variability.
>
> While our study focuses on affine deformations, the underlying mathematical framework directly extends to more general mappings, including diffeomorphic representations. Recent work (e.g., [1] Yin et al.) illustrates how parameterized domains can be extended to PCA-based and fully diffeomorphic geometries, suggesting a clear path for future work.
>
> To address the reviewer’s concern, we have added an anisotropic 2D experiment in Section 6 (Table 3 and Figure 2), demonstrating that the latent PDE mapping technique accommodates more complex, direction-dependent PDE dynamics. We also extend our geometric representation beyond affine parameters by incorporating PCA-based inputs in Appendix E.1.1 (Table 10). These additions show that LPM-PINN generalizes to more complex PDE behavior and can be readily adapted to non-affine geometric parameterizations.
>
> [1] Yin M, Charon N, Brody R, Lu L, Trayanova N, Maggioni M. A scalable framework for learning the geometry-dependent solution operators of partial differential equations. Nature computational science. 2024 Dec;4(12):928-40.
>
> **[W2] Limited ablation on: the necessity of the full deformation-gradient-based mapping vs simpler coordinate transforms, sensitivity to the choice of latent domain, computational overhead trade-offs.**
>
> We agree that a more complete analysis of sensitivity and computational overhead would strengthen the work. In the revised version, we have added Appendix G, which reports training times (Table 11), inference times (Table 12), and the one-time cost of computing deformation gradients and mappings. These additions clarify the trade-offs of the proposed framework and provide a more complete picture of its practical feasibility. We have also expanded the discussion in Section 7 to highlight the computational considerations.

---

### Author Response · Authors · 2025-11-22

We would like to sincerely thank all reviewers for their thoughtful assessments and constructive feedback. We appreciate the time and effort invested in evaluating our work. In response to the comments, we have uploaded a revised version of the paper that includes the following updates:

- **Expanded introduction and related work:** We now more clearly frame the sparse-data regime targeted in this study and explain why a physics-informed approach is necessary compared to purely data-driven methods.
- **Clarification of combined geometry families:** Additional details describing how the combined geometry families were constructed have been added to Section 5.
- **New 2D anisotropic experiment:** We have included an anisotropic diffusion experiment to Section 6 (Table 3 and Figure 2) to demonstrate that the LPM technique extends to more complex PDE dynamics.
- **Unified snapshot times:** All figures now use consistent snapshot times to facilitate clearer comparisons across methods.
- **Updated PDE parameter information:** Appendix D has been expanded to include the previously missing PDE parameters in Table 8, and additional details regarding the diffusion tensor D and conduction values (Table 9).
- **Computational analysis:** We have included an additional appendix (Appendix G) to provide insight to the training times (Table 11), inference times (Table 12) and an estimation of the computational overhead related to deformation gradient calculations and mapping to a reference geometry.
- **DeepONet implementation and results:** A new appendix (Appendix H) describes our implementation of DeepONet and reports results on the 2D isotropic experiments. Although this comparison is not directly aligned with our sparse-data focus, we include it to verify empirically that neural operator approaches struggle in this regime, reinforcing the motivation for a physics-informed method.

We hope these revisions address the reviewers’ concerns and help clarify the contributions and scope of our work.

---

### Comment · Area_Chair_qK8i · 2025-11-26
**Please Review Author Response**

Dear Reviewers,

The authors have now responded to your comments. Could you review their response as soon as possible? If you have any further questions or concerns, please raise them as well.

Best,

Your AC

---

### Meta-Review · Area_Chair_8M9j · 2026-01-02

**Summary:**

The paper received mixed scores, and the majority of reviews before the rebuttal period were negative. The reviews raised three major concerns in their initial reviews:

1) The problem setting demonstrated in the experiments is limited to simplified PDE dynamics and affine transformation. Most reviewers considered this a limitation or weakness.
2) Many reviewers suggested a more comprehensive analysis of the method, including comparing it with more baselines, reporting more detailed computational cost, conducting more ablation studies, and testing on out-of-range cases.
3) Some reviewers questioned the paper’s technical novelty and contribution over prior works in both numerical methods (e.g., finite element methods) and learning-based approaches (e.g., some variants of neural operators)

Reviewers also raised individual concerns regarding the paper’s storytelling, method designs, and potential applications.

**Reviewer Concerns:**

The rebuttal responded to the first concern by 1) clarifying the technical merit of their synthetic PDE problem in providing a clear and controllable test bed, 2) adding a new, anisotropic 2D PDE problem in their experiments, and 3) extending their geometric representation beyond affine transformation. I think the rebuttal has largely addressed this concern.

Regarding the second concern, the rebuttal reported more performance statistics to reflect its time cost and added DeepONet as a baseline in the appendix. I think this effort helps demonstrate the method’s efficacy and better position it with respect to relevant works (e.g., neural operators). Still, I am unsure whether reviewers would consider the new DeepONet experiment sufficient, as it was conducted on 2D isotropic test problems.

Regarding the last concern, the rebuttal stressed that the paper intentionally focused on a sparse-data regime, which differs from those considered in the cited papers discussed by the reviewers. I think both the reviews and the rebuttal have some fair points here. I think this concern would require a few more rounds of discussion before being fully resolved.

**Reviewer Scores:**

The positive review (6) was fairly short. I think the reviewer would likely maintain their positive score after full discussion.

The borderline negative (4) review is associated with modest confidence (3). It was also fairly short and expressed concerns similar to those in other reviews. I think the reviewer might become a borderline supporter if all other reviewers were supportive.

The remaining two reviewers were quite negative (2) and with high confidence (5). I think it would be difficult to convince them to show more support for the paper, as their major concerns span across multiple aspects of the paper, requiring more complex test problems and more technical novelty over existing approaches.

---

### Decision · Program_Chairs · 2026-01-26

Reject